*Auto-Annotation from Expert-Crafted Guidelines*:
# A BENCHMARK THROUGH 3D LiDAR DETECTION

## ABSTRACT

A crucial yet under-appreciated prerequisite in machine learning solutions for real-applications is data annotation: human annotators are hired to manually label data according to expert-crafted guidelines. This is often a laborious, tedious, and costly process. To study methods for automated data annotation based on expert-crafted annotation guidelines. we introduce a new benchmark *AutoExpert*, short for *Auto-Annotation from Expert-Crafted Guidelines*. In particular, this work repurposes the well-established nuScenes dataset, commonly used in autonomous driving research, which provides comprehensive annotation guidelines for labeling LiDAR point clouds with 3D cuboids across 18 object classes. In the guidelines, each class is defined by multimodal data: a few visual examples and nuanced textual descriptions. Notably, no labeled 3D cuboids in LiDAR are provided in the guidelines. The clear discrepancy between data modalities makes AutoExpert not only challenging but also novel and interesting. Moreover, the advances of foundation models (FMs) make AutoExpert especially timely, as FMs offer promising tools to tackle its challenges. To address AutoExpert, we employ a conceptually straightforward pipeline that (1) utilizes open-source FMs for object detection and segmentation in RGB images, (2) projects 2D detections into 3D using known camera poses, and (3) clusters LiDAR points within the frustum of each 2D detection to generate a 3D cuboid. Starting with a non-learned solution that leverages off-the-shelf FMs, we progressively refine key components and achieve significant performance improvements, boosting 3D detection mAP from 12.1 to 21.9. Nevertheless, AutoExpert remains an open and challenging problem, underscoring the urgent need for developing LiDAR-based FMs.

## 1 INTRODUCTION

Foundation Models (FMs) have emerged as powerful backbones that can significantly boost performance on downstream tasks when fine-tuned with meticulously annotated task-specific data (Shen et al., 2025; Liu et al., 2025). Data annotation remains a critical yet costly prerequisite for many real-world machine learning applications. It essentially starts with expert-crafted annotation guidelines, which serve as the basis for training human annotators (Caesar et al., 2020). Given the importance of annotation guidelines and the rapid advancements of FMs, we leverage FMs to investigate the task of *Auto-Annotation from Expert-Crafted Guidelines (AutoExpert)*: automatically labeling data based on annotation guidelines crafted by domain experts. In particular, we explore AutoExpert in the context of 3D detection (see Fig. 1). Addressing the AutoExpert task not only offers practical value in reducing annotation costs but also provides a compelling benchmark for assessing how effectively current FMs can facilitate solving highly specialized real-world problems.

**Status quo.** A variety of machine learning research problems are proposed to reduce the cost of data labeling, such as active learning (Holub et al., 2008; Settles, 2009; Kirsch et al., 2019; Ren et al., 2021; Bang et al., 2024) and few-shot learning (FSL) (Snell et al., 2017; Boudiaf et al., 2020; Bateni et al., 2020; Satorras & Estrach, 2018). Active learning leverages the model being trained to selectively identify the most informative unlabeled samples for annotation, under the assumption that human annotators are already trained and familiar with the labeling policies. In contrast, FSL aims to train models using only a small number of labeled examples. Recent studies begin to explore FSL from the perspective of data annotation (Madan et al., 2024; Liu et al., 2025), using annotation guidelines that include few-shot visual examples per class. However, most existing FSL works

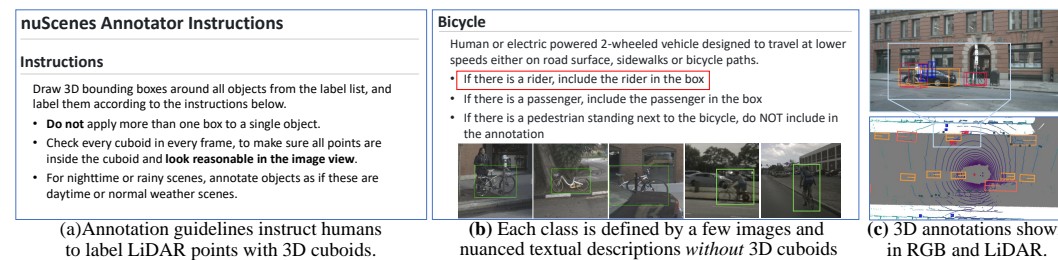

(a)Annotation guidelines instruct humans to label LiDAR points with 3D cuboids. **(b)** Each class is defined by a few images and nuanced textual descriptions *without* 3D cuboids **(c)** 3D annotations shown in RGB and LiDAR.

Figure 1: **Screenshots of annotation guidelines released by the nuScenes dataset** (Caesar et al., 2020). **(a)** The guidelines instruct human annotators to label LiDAR points with 3D cuboids for specific object classes. **(b)** Each class is defined with a few visual examples and nuanced textual descriptions (ref. the red box) *without* 3D annotations. Human annotators must interpret and apply these guidelines to manually generate 3D cuboids. **(c)** We visualize the 3D annotations in both RGB images and the Bird's-Eye-View (BEV) of the LiDAR point cloud.

focus on relatively simple vision tasks such as image classification (Liu et al., 2025) and object detection (Madan et al., 2024), typically relying on a single FM. To advance the study of automated annotation from guidelines, we introduce this new benchmark *AutoExpert*.

**Benchmark.** We construct the AutoExpert benchmarking protocol through the lens of 3D detection in LiDAR data. Specifically , we repurpose the established nuScenes dataset (Caesar et al., 2020) which releases annotation guidelines designed to train human annotators (Fig. 1). These guidelines define each object category using only a few visual examples and textual descriptions, *without* 3D cuboid annotations. Therefore, AutoExpert can be framed as *multi-modal few-shot learning for 3D detection without 3D annotation*. Compared to standard FSL tasks, AutoExpert is significantly more challenging. In particular, it calls for leveraging Large Language Models (LLMs) to interpret textual annotation guidelines, Vision-Language Models (VLMs) to detect objects defined by textual descriptions, and Vision Foundation Models (VFMs) for object detection and segmentation. By combining these capabilities, AutoExpert serves as valuable testbed for evaluating how well diverse FMs can be applied to complex, real-world annotation tasks.

**Challenges.** AutoExpert presents unique and interesting challenges. First, its goal is 3D detection in LiDAR data but the supervision available for training comes from a few visual examples and textual descriptions without 3D annotations (see Fig. 1). Translating such high-level instructions into actionable supervision for machines to learn to annotate point clouds with 3D cuboids is non-trivial. Second, while it might be tempting to leverage open-source FMs, there are currently no publicly-available LiDAR-based FMs. This makes it challenging to directly apply existing FMs to LiDAR-based 3D detection. Third, given that the nuScenes annotation guidelines include both visual examples and textual descriptions, a natural formulation is to cast AutoExpert as a *multimodal few-shot learning* problem: leveraging multiple FMs to utilize visual and textual guidance for specialized 3D detection. Yet, developing an effective solution by leveraging proper FMs to address such a highly-specialized task remains an open problem and under-explored in the literature.

**Methodology.** By addressing the challenges outlined above, we present a conceptually straightforward pipeline (Fig. 2) to tackle AutoExpert. Our approach integrates multiple FMs to (1) interpret annotation guidelines, (2) prompt foundational 2D detectors, and (3) segment object instances in RGB images. Using synchronized LiDAR and camera sensors, we lift detected object from 2D RGB frames into 3D space and generate 3D cuboids in the LiDAR point cloud as 3D detections. Further, we progressively refine key components within the pipeline, greatly boosting 3D detection performance. Despite these improvements, our comprehensive analysis shows that AutoExpert remains far from a solved problem. We evaluate each core component and offer insights to inspire future research.

**Contributions.** Our paper is positioned as a benchmark paper, making three key contributions:

1. We introduce a novel and timely task, *AutoExpert*, which not only promotes the development of practical data annotation methods but also facilitates evaluation of FMs in real-world scenarios.

2. We present a benchmarking protocol for studying AutoExpert by repurposing the well-established nuScenes dataset. Our benchmark includes code, data, metrics, and a suite of baseline models.

3. We address AutoExpert with a conceptually simple yet effective pipeline that integrates multiple FMs for 2D detection, segmentation, and 3D detection. We improve key components of the pipeline, greatly boosting performance and offering valuable insights to guide future research.

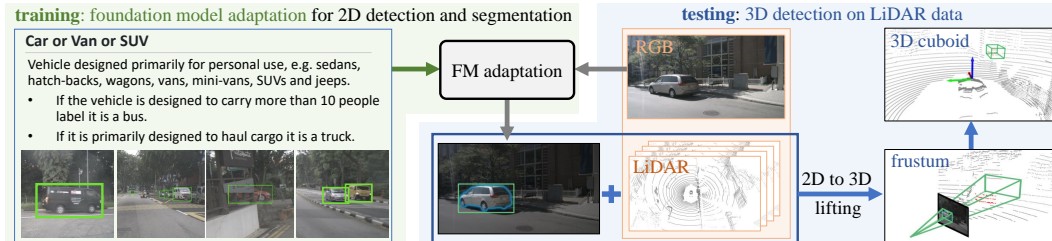

Figure 2: We present a pipeline to solve AutoExpert by adapting open-source foundation models (FMs). Specifically, over the visual examples and textual descriptions that define object classes of interest, we adapt a Vision-Language Model (VLM) and a Vision Foundation Model (VFM) for object detection and segmentation. The adapted FMs produce decent 2D detections on unlabeled RGB frames. With the aligned LiDAR and RGB frames, we lift each 2D detection to 3D, locate corresponding LiDAR points, and generate a 3D cuboid as the 3D detection. We propose simple and novel methods to improve FM adaption and 3D cuboid generation (Section 4).

## 2 RELATED WORK

**Data Annotation** is a critical yet costly prerequisite for machine learning solutions in numerous real-world applications. Abundant works are motivated to reduce annotation cost with human-in-the-loop (Abad et al., 2017; Heo et al., 2020; Cheng et al., 2021; Qiao et al., 2023) or active learning (Holub et al., 2008; Settles, 2009; Kirsch et al., 2019; Ren et al., 2021; Bang et al., 2024). But these works over-simplify the complexity of data annotation by focusing on general object categories (e.g., `car` and `person`) (Ramanan & Forsyth, 2003; Reza et al., 2025; Wu et al., 2024a; Wang et al., 2006), without considering real-world applications that require labeling task-critical classes defined with nuanced details (Madan et al., 2024). For instance, for autonomous driving safety, `bicycle` is defined differently (Caesar et al., 2020) from what one would have thought (Fig. 1). Even the recent work (Zhou et al., 2024), which studies annotation in LiDAR with RGB and VLM, also focuses on general object classes such as `car` and `motorcycle`. Hence, it remains an open question whether one can develop methods to automatically annotate domain-specific data directly from expert-crafted annotation guidelines. Our work explores this new problem AutoExpert.

**Few-Shot Learning** (FSL) aims to develop methods to learn from a small number of labeled examples (Snell et al., 2017; Boudiaf et al., 2020; Bateni et al., 2020; Satorras & Estrach, 2018). Recent FSL methods propose to adapt a pretrained VLM (Zhang et al., 2022b; 2023c; Lin et al., 2023; Tang et al., 2024; Silva-Rodriguez et al., 2024; Khattak et al., 2023; Liu et al., 2025). Further, some recent works point out that FSL is better studied from a data annotation perspective, as annotation guidelines contain few-shot visual examples and textual descriptions (Madan et al., 2024; Liu et al., 2025). However, the current FSL literature has largely focused on simple tasks such as image classification (Madan et al., 2024) and object detection (Liu et al., 2025), and typically employs a single FM. In contrast, our AutoExpert introduces a more challenging setting that requires developing multi-modal few-shot learning methods for 3D LiDAR detection without 3D annotation. Beyond its technical challenges, AutoExpert also holds practical significance as its evaluation protocol is grounded in real-world annotation practices, making use of authentic, official annotation guidelines.

**Foundation Models (FMs)** are core to today's leading AI products such as GPT-4o (Achiam et al., 2023), Gemini (Team et al., 2023) and Qwen (Bai et al., 2023). As our work focuses on leveraging open-source Vision-Language Models (VLMs) and Visual Foundation Models (VFMs), we briefly review them. VLMs are pretrained on large-scale image-text pairs (Radford et al., 2021; Jia et al., 2021; Liu et al., 2023a;b; Bai et al., 2023), achieving unprecedented results in visual understanding tasks such as visual grounding, image captioning, and visual question answering. VFMs, by contrast, are trained primarily on visual data (Chen et al., 2020; Caron et al., 2021; Zhou et al., 2022; Oquab et al., 2023; Touvron et al., 2022) and excel at perception tasks such as object detection (Zhang et al., 2022a; Liu et al., 2023b; Ren et al., 2024; Liu et al., 2023b) and segmentation (Kirillov et al., 2023; Wang et al., 2024). Recent efforts have attempted to transfer the general perception abilities of VLMs to LiDAR perception by associating VLM output with LiDAR points through geometric alignment between camera and LiDAR sensors (Khurana et al., 2024; Osep et al., 2024). However, such approaches fall short in domains like autonomous driving, where object definitions require nuanced understanding critical to safety (Madan et al., 2024). Therefore, adapting FMs effectively to address real-world, task-specific challenges is essential (Madan et al., 2024; Liu et al., 2025). Our work evaluates FMs through the highly-specialized yet widely-studied task: 3D LiDAR detection.

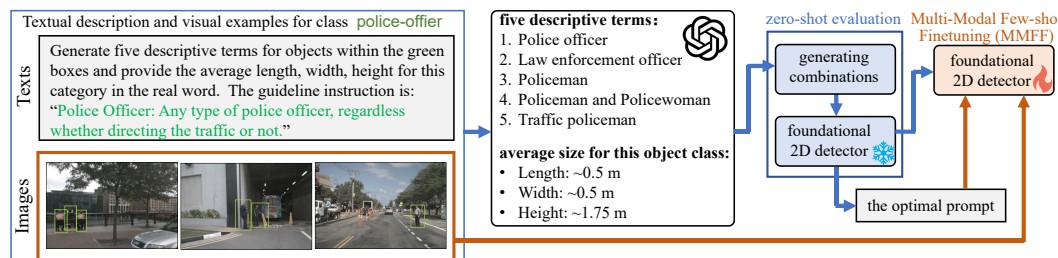

Figure 3: For each class name, we use a pretrained VLM (GPT-4o (Achiam et al., 2023) in this work) to find a list of terms that match the textual description and visual examples provided in the annotation guidelines. We select the term or its combination that maximizes the zero-shot detection performance of a foundational object detector (GroundingDINO (Liu et al., 2023b) in this work) on the validation set. We use the selected terms to finetune the detector, yielding notable improved performance (Table 1).

**Foundation Model Evaluation** is usually done through general tasks such as mathematical reasoning (Lu et al., 2023), visual understanding (Wu et al., 2023) and language reasoning (Zhong et al., 2023). Our AutoExpert benchmark extends this evaluation to 3D perception in the context of autonomous driving, i.e., 3D LiDAR detection, motivated from the perspective of real-world data annotation practices. To the best of our knowledge, there is currently no FM specifically designed for LiDAR-based perception. Therefore, the AutoExpert benchmark evaluates the performance of proper combinations of existing FMs. Through AutoExpert, our extensive experiments reveal significant room for improving in FM adaptation for 3D LiDAR detection.

**3D LiDAR Detection** has been extensively studied in autonomous driving research, leading to the release of several large-scale datasets such as nuScenes (Caesar et al., 2020), KITTI (Geiger et al., 2013), Waymo (Sun et al., 2020), and Argoverse2 (Wilson et al., 2023). Among them, nuScenes is the only one that released its *official annotation guidelines*. On the contrary, others released user guides to help challenge participants to get familiar with their data. To approach 3D LiDAR detection, most methods train 3D detectors over massive annotated LiDAR data, optionally with annotated RGB frames (Yin et al., 2021; Bai et al., 2022; Li et al., 2024); some explore training 3D LiDAR detectors in an unsupervised manner (Zhang et al., 2023b; Wu et al., 2024b). Notably, till now, these methods focus on common object classes (e.g., car and cyclist) and neglecting rare but safety-critical ones (e.g., stroller and wheelchair), although annotation guidelines have defined all such classes (Peri et al., 2023). In contrast, our AutoExpert benchmark evaluates on all these classes. Further, per annotation guidelines, AutoExpert does not provide 3D annotations as training data.

## 3   AUTOEXPERT: PROBLEM FORMULATION AND BENCHMARKING PROTOCOL

**Problem Formulation.** In plain terms, AutoExpert mimicks human annotators to label LiDAR data using 3D cuboids. As the annotation guidelines (Fig. 1) contain only textual descriptions and a few 2D visual examples without 3D cuboids references, any developed methods must learn from the visual and textual information to generate 3D cuboids on the LiDAR data. Mirroring human annotators' workflow, the developed methods are expected to (1) understand each object class with the help of textual descriptions and visual examples, (2) detect objects in RGB frames and associate LiDAR points to them, (3) utilize prior knowledge about objects' 3D shapes and sizes to generate appropriate 3D cuboids in the LiDAR point cloud. We evaluate methods primarily on 3D LiDAR detection quality; we also use 2D metrics to assess methods that generate 2D detections as intermediate outputs.

**Data preparation.** We repurpose the nuScenes dataset (Caesar et al., 2020) which is publicly available under the CC BY-NC-SA 4.0 license. The dataset provides annotations for 18 object classes. While its official annotation guidelines contain images to demonstrate each class (Fig. 1), we do not use them in our benchmark as these images are non-nuScenes images and are likely sourced from the Internet that potentially have copyright concerns. Therefore, we replacing them with 4-8 selected nuScenes images per class from the official training set. These selected images clearly capture visual signatures of objects, simulating iconic visual examples displayed in annotation guidelines. Importantly, we adhere to annotation guidelines that exclusively demonstrate each class with visual examples overlaid with annotations only for that class. Therefore, for each selected image, we retain only the annotations of the target class and discarding those belonging to other classes. For example,

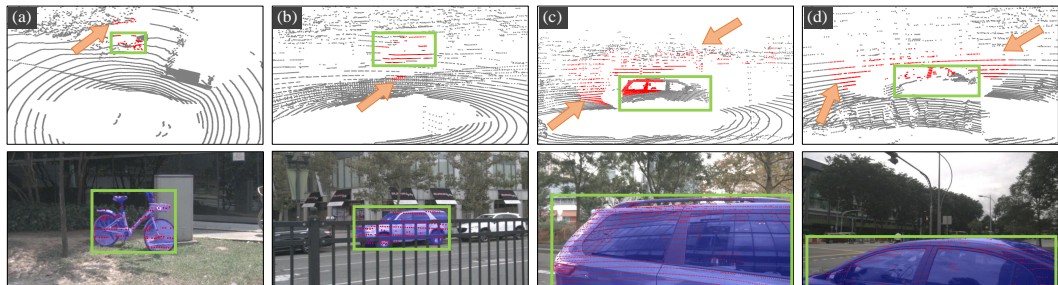

Figure 4: Generating a 3D cuboid based on LiDAR points is challenging as points can be from occlusders and backgrounds. For example, (a) LiDAR points projected on a `bicycle` can be from background through wheels; (b) points projected on a `car` can be from a fence occluding the car; (c-d) points projected on a `car` can be background through the windows and windshield of the car.

Fig. 3 shows selected frames for `police officer` class, where objects like `car` and `traffic cone` are present but not annotated.

**Train, validation, and test sets**. We consider the images and textual descriptions in the annotation guidelines as our training set. Notably, AutoExpert follows the nuScenes benchmark to focus on 3D detection in LiDAR data but the nuScenes annotation guidelines do not provide 3D cuboid annotations for demonstration. Therefore, our training set does not contain annotated LiDAR data. Moreover, from the nuScenes' official validation set (containing 6,019 annotated LiDAR sweeps and 36,114 annotated RGB frames), we sample 570 frames as our validation set for hyperparameter tuning and model selection. This small validation set simulates the expert-in-the-loop quality control, where domain experts typically oversee annotation progress and quality. We use the nuScenes' official validation set as our test set. We release the data splits in this anonymous GitHub repository.

**Metrics.** We evaluate methods w.r.t both 2D detection and 3D detection metrics:

- *2D metrics.* Following (Lin et al., 2014), we report mean Average Precision which is the mean of per-class AP at IoU=0.5. We denote this metric as $\text{mAP}^{2D}$.

- *3D metrics.* Following nuScenes (Caesar et al., 2020), we first report mean Average Precision over per-class AP at different ground-plane distance thresholds, [0.5, 1.0, 2.0, 4.0] in meters. We denote this metric as $\text{mAP}^{3D}$. We also report the nuScenes Detection Score (NDS), which summarizes translation error, scale error, orientation error, velocity error, and attribute error. Appendix E provides more results w.r.t these specific metrics.

## 4 METHODOLOGY: LIFTING 2D DETECTIONS TO 3D

To address and understand the challenges of the new problem AutoExpert, we present an intuitive framework in which we leverage proper FMs rather than proposing sophisticated methods, as illustrated in Fig. 2. The pipeline has two key components: (1) 2D object detection on RGB frames, and (2) 3D cuboid generation for 2D detection. We describe them with our methods in Section 4.1 and 4.2 respectively, and a few more technical enhancements in Section 4.3.

### 4.1 2D DETECTION BY MULTI-MODAL FEW-SHOT FINETUNING

For 2D detection on RGB frames, we exploit the open-source foundational detector GroundingDINO (Liu et al., 2023b). Foundational detectors yield impressive zero-shot detection performance on natural images but they are not tailored to specific tasks (Madan et al., 2024). For example, for autonomous driving as in nuScenes, `bicycle` is defined differently from commonsense (Fig. 1): annotators should include the existing rider in the box annotation for driving safety concerns. We improve GroundingDINO with novel techniques described below.

**Prompt Engineering** designs prompts that can lead to better zero-shot performance (Parashar et al., 2023; 2024). This is often done manually, but in this work, we present an *automated* method to generate better prompts for object detection (Fig. 3). Specifically, for each object class, we prompt a VLM to find five descriptive terms that fit the textual descriptions and visual examples provided in the guidelines. Then, we test the term and their combinations to prompt GroundingDINO for zero-shot

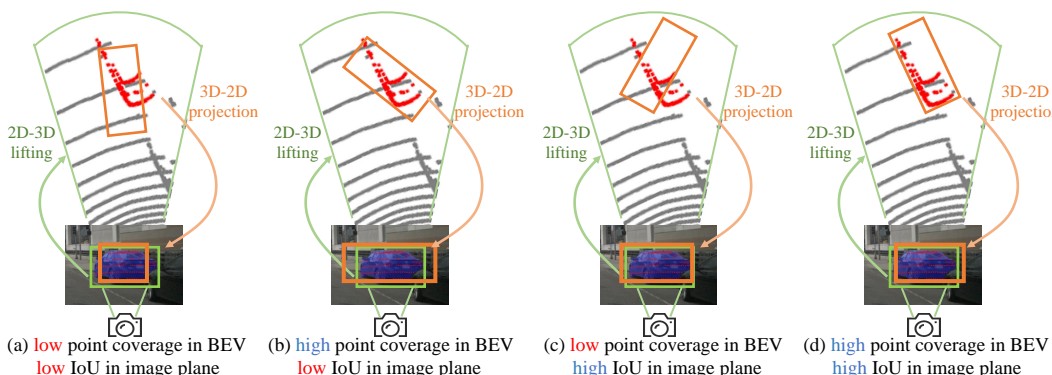

(a) low point coverage in BEV low IoU in image plane
(b) high point coverage in BEV low IoU in image plane
(c) low point coverage in BEV high IoU in image plane
(d) high point coverage in BEV high IoU in image plane

Figure 5: For each 2D detection on an image, we generate a 3D cuboid via Multi-Hypothesis Testing (MHT). Specifically, in the frustum projected from the 2D detection, we locate the LiDAR points that lie within the mask of the detected object once projected to the image plane. Next, for the detected object (the `car` displayed as an example), we use the size prior obtained by prompting GPT-4o (ref. Fig. 2) to fit the LiDAR points in 3D. We pre-define a list of rotation and translation step sizes to measure their point coverage in BEV and IoU on the image plane, and find the best-fitting one that yields the highest point coverage and IoU. Despite its simplicity, our MHT method outperforms previous approaches of zero-shot 3D LiDAR detection and self-supervised learning for 3D LiDAR proposal detection (Table 2).

detection. We pick the terms that yield the highest 2D detection precision on the validation set. We use the selected terms for all the classes to prompt GroundingDINO for zero-shot object detection.

**Multi-Modal Few-Shot Finetuning.** As a small amount of images are available in the annotation guidelines, we use them as few-shot training examples (which have box annotations as ground-truth labels) to finetune the foundational detector (Fig. 3). It is worth noting visual examples in the guidelines are annotated in a federated way, i.e., all objects belonging to the focused class are annotated while others are not. Therefore, when finetuning on the few-shot images, we compute the loss over training images pertaining to the specific classes without treating detections of other classes as false positives. Importantly, finetuning with the selected terms (as presented in the previous paragraph) performs better than original class names (Table 1).

## 4.2 3D Cuboid Generation by Multi-Hypothesis Testing

For each detection box in the 2D image obtained above, we construct a frustum using camera parameters and identify LiDAR points therein (ref. bottom-right of Fig. 2). To generate a proper 3D cuboid for this detection, we design a baseline that uses size prior based on the predicted class (ref. Fig. 3) to determine the size of the 3D cuboid. Below, we describe how to place this 3D cuboid.

**Background Points Removal via Foreground Segmentation.** LiDAR points within the detection box may include background points, particularly near the box boundaries. To address this, we perform foreground segmentation to precisely segment the object within the detection box. We use the foundational segmentation model SAM (Kirillov et al., 2023) by prompting it with the 2D detection box. Fig. 4 and 5 display decent segmentation results by SAM, indicating the effectiveness in refining LiDAR point selection. In fact, previous works have explored LiDAR points and foreground segmentation to lift 2D detections to 3D (Wu et al., 2024a; Khurana et al., 2024; Zhou et al., 2024) but have not address notable critical challenges — the remaining LiDAR points can be from occluders (e.g., a fence in front of a vehicle) and background artifacts (e.g., points on a wall visible through a vehicle's windshield and windows), as shown in Fig. 4. Below, we present a method to mitigate these issues via Multiple Hypotheses Testing (MHT) (Shaffer, 1995).

**3D Cuboid Generation via MHT.** Following the nuScenes annotation guidelines, human annotators manually fit 3D cuboids to LiDAR points belonging to objects identified in 2D images. We automate this process through an MHT-based approach (Fig. 5). For a 2D detection, we first initialize a cuboid that covers LiDAR points within the frustum. We determine its dimensions based on its identified class by the 2D detector and the size prior obtained from GPT-4o (Fig. 3). We translate and rotate this cuboid within the frustum w.r.t predefined step sizes. We select the most desirable cuboid that has (1) maximal coverage of LiDAR points within the cuboid, and (2) highest Intersection-over-Union (IoU)

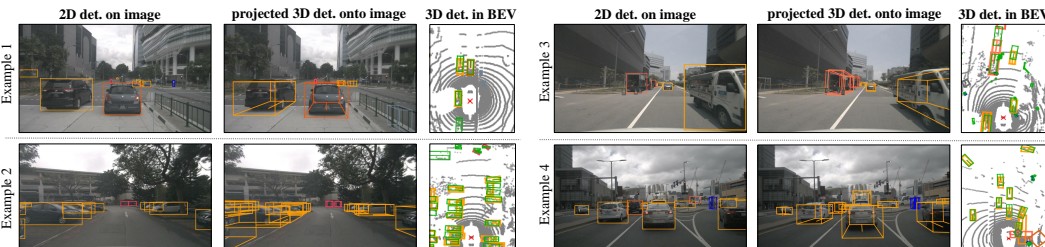

Figure 6: **Visualization of detection results on four testing examples.** For each example, we display 2D detections, and 3D detections (i.e., the generated 3D cuboids) projected onto the RGB image and the BEV of LiDAR data. Results show that our method decently detects objects that are in far field and small in size, which are usually challenging to detect (Peri et al., 2023; Gupta et al., 2023). Appendix F contains more visual results.

between the original 2D detection box and the projected 2D box of the 3D cuboid. We discuss the time cost of this MHT method and provide implementation details in Appendix D.

### 4.3 TECHNIQUES FOR PERFORMANCE ENHANCEMENT

We present a few more techniques to improve 3D cuboid generation, elaborated below.

**LiDAR Sweep Aggregation.** As a single LiDAR sweep can be too sparse to precisely localize objects in 3D, we aggregate multiple sweeps. While existing 3D detection methods have explored aggregating *history* sweeps at a given timestamp, AutoExpert additionally allows aggregating "*future*" sweeps to facilitate data annotation. We analyze per-category 3D detection performance w.r.t different aggregation strategies (Table 3) and find that aggregation strategies significantly impact detection accuracy for certain classes, depending on the object size and moving speed of each class.

**3D Cuboid Scoring with Geometric Cues.** We score each generated 3D cuboid using both the 2D detection confidence $S_{2D}$ and 3D geometric information. To capture 3D geometric cues, we compute an occupancy rate (Wu et al., 2024b) based on LiDAR point distribution within the cuboid. Specifically, we (1) project the 3D cuboid onto the ground plane, obtaining a BEV rectangle, (2) discretize this rectangle into a 7×7 grid, (3) count grid cells ($N$) that contain at least one project LiDAR point, and (4) compute the occupancy rate as $S_{3D} = N/49$. The final cuboid score combines these metrics through weighted summation: $S_{2D}$ and $S_{3D}$, i.e., $S = \alpha * S_{2D} + (1 - \alpha) * S_{3D}$, where the weighting parameter $\alpha$ is optimized to maximize 3D detection precision on the validation set.

**Tracking-Based Score Refinement.** To refine the scores of generated 3D cuboids at a given timestamp, we leverage temporal information by "tracking" objects across LiDAR sweeps. Concretely, we adopt a heuristic approach to associate 3D cuboids which have the same class labels (predicted by the 2D detector) and are spatially close. For all the 3D cuboids belong to the same track, we replace their individual scores with their mean score. Despite its simplicity, this method remarkably improves 3D detection performance (Table 4).

## 5 EXPERIMENTAL RESULTS AND ANALYSIS

We conduct a series of experiments to validate the effectiveness (Fig. 6) of our methods and demonstrate the challenges in AutoExpert. We start by introducing important implementation details.

**Implementations.** FMs exploited in this work contain the GPT-4o (Achiam et al., 2023) for annotation guidelines understanding, the VLM GroundingDINO as a 2D detector (Liu et al., 2023b), and the VFM SAM (Kirillov et al., 2023) for object segmentation. When operating with FMs, we use an NVIDIA A6000 GPU. We use Python and PyTorch in experiments. We provide more details about FM adaptation in Appendix L and release our code and models anonymously at GitHub.

**Analysis on 2D detection.** We analyze 2D detection performance of different finetuning methods with GroundingDINO. We compare these finetuning methods in Table 1: (1) using the original class names to prompt GroundingDINO for zero-shot detection (dubbed "GD w/ o-name"), (2) finetuning GroundingDINO on the few-shot examples with the original class names (dubbed "ft-GD w/ o-name"), (3) using GPT-4o refined class names to prompt GroundingDINO (dubbed "GD w/ r-name"), and (4) finetunes GroundingDINO on the few-shot examples with the GPT-4o refined

Table 1: Different 2D detector finetuning methods yield different 3D detection performance. Here, the 2D detector is GroundingDINO (GD). To report 3D metrics, we generate 3D cuboids for 2D detections using CM3D (Khurana et al., 2024). Recall that we refine class names using an off-the-shelf foundation model (GPT-4o as illustrated in Fig. 3). When prompting GD for zero-shot detection, using refined class names ("r-name") performs better than using the original class names ("o-name"). Importantly, finetuning GD (ft-GD) using r-name performs the best. The Appendix (Table 7 & 10) presents more results.

|  | GD o-name | ft-GD o-name | GD r-name | ft-GD r-name |
|---|---|---|---|---|
| $mAP^{2D}$ | 16.9 | 20.0 | 18.2 | **20.8** |
| $mAP^{3D}$ | 12.1 | 16.6 | 15.7 | **18.2** |
| NDS | 16.6 | 21.2 | 22.1 | **23.1** |

Table 2: Comparison of methods for 3D detection, i.e., 3D cuboid generation. Here, we improve the compared methods Oyster, LISO and CPD with box fitting (Zhang et al., 2017; You et al., 2022) which refines their 3D detections. Results demonstrate that our MHT method achieves superior performance in both $mAP^{3D}$ and NDS metrics Refer to Table 10 in the Appendix for more metrics.

| Method | $mAP^{3D}$ | NDS |
|---|---|---|
| SAM3D (Zhang et al., 2023a) | 1.6 | 6.9 |
| Oyster (Zhang et al., 2023b) | 6.3 | 10.7 |
| Oyster w/ frustum | 12.9 | 15.6 |
| LISO (Baur et al., 2024) | 8.9 | 13.1 |
| LISO w/ frustum | 15.7 | 19.8 |
| CPD (Wu et al., 2024b) | 10.1 | 14.2 |
| CPD w/ frustum | 17.9 | 22.3 |
| CenterPoint (Yin et al., 2021) | 3.6 | 19.0 |
| CM3D (Khurana et al., 2024) | 18.2 | 23.1 |
| MHT | **19.2** | **23.8** |

class names (dubbed "ft-GD w/ r-name"). Interesting, finetuning with refined class name ("ft-GD w/ r-name") performs the best: it finds that changing class names such as police-officer to law enforcement officer facilitate FM adaptation. Refer to Appendix B for more details.

**Comparison of 3D LiDAR detection methods.** We compare our MHT-based 3D cuboid generation (Section 4.2) against various existing 3D detection approaches, spanning zero-shot 3D detectors, self-supervised 3D proposal detectors, and a 3D detector supervised-learned on another dataset.

- SAM3D (Zhang et al., 2023a) is a zero-shot 3D detector that exploits SAM (Kirillov et al., 2023) to segment objects on BEV images of LiDAR data; it struggles to detect small objects.
- Oyster (Zhang et al., 2023b) and CPD (Wu et al., 2024b) are unsupervised learned models that were developed and benchmarked on Waymo dataset (which contains only three classes, vehicle, pedestrian and cyclist); they struggle to detect diverse object classes.
- LISO (Baur et al., 2024) is a self-supervised LiDAR 3D detector on movable objects without class label prediction. Note that LISO, Oyster and CPD adopt DBSCAN clustering for 3D box fitting (Zhang et al., 2017; You et al., 2022).
- We also compare the 3D detector CenterPoint (Yin et al., 2021) supervised trained on Argoverse2, intending to demonstrate the significant gap between LiDAR models (see details in Appendix C).

As many of these methods focus on producing 3D cuboids without predicting class labels, we assign class labels to their 3D detections: first matching their 3D detections with GroundingDINO's 2D detections, then assigning GroundingDINO's predicted classes to the matched 3D detections. Based on these 3D proposal approaches, we design their variants by using GroundingDINO: we use GroundingDINO's 2D detections to not only offer class labels but also define frustums, in which we run these approaches to produce 3D detections. The frustums provide more targeted searching space that can improve these methods. We append "w/ frustum" to denote these variants. Refer to Appendix E for more details of these methods. Table 2 compares their results on 3D detection, demonstrating the superior performance of our MHT-based method. Moreover, our modifications (marked with "w/ frustum") for the compared methods notably improves their performance. Interestingly, CenterPoint achieves quite poor performance, although we carefully tuned hyperparameters and unified Argoverse2 LiDAR data format with nuScenes. We find the reason is due to the significant gap between LiDAR sensors in the two datasets (details in Appendix C). This demonstrates a need of developing LiDAR-based foundation models.

**Analysis of LiDAR sweep aggregation.** We report per-class detection precision by applying different aggregation strategies. Table 3 shows that different aggregations notably improve on certain classes (refer to Table 14 in Appendix for the full results). Interestingly, results are "asymmetric", e.g., for traffic-cone, construction-worker and bicycle, aggregating the future two sweeps yields significantly better performance than aggregating the past two sweeps; for child, aggregating the past 6 sweeps is significantly better than other strategies! We conjecture the reason is due to objects' motion speed and patterns. It is worth noting that on typical "rare" classes, our method even outperforms the state-of-the-art supervised approach (Peri et al., 2023), which report 3.4 mAP on child, whereas ours achieves 5.1 mAP.

Table 3: **Analysis of sweep aggregation strategies on per-class 3D detection performance** (mAP$^{3D}$). "$P$+C+$F$" denotes aggregating the past $P$ sweeps, the current sweep C, and the future $N$ sweeps; we drop $P$ or $F$ if not aggregating any past or future sweeps. In each row, we bold the highest number. Somewhat surprisingly, aggregation strategies greatly impacts performance on certain classes, e.g., for `construction-worker`, `bicycle` and `traffic-cone`, aggregating the future 2 sweeps yields remarkably better performance than others. Refer to Table 14 in Appendix for the full results.

| Class | 10+C | 6+C | 2+C | C | C+2 | C+6 | C+10 | 1+C+1 | 3+C+3 | 5+C+5 |
|---|---|---|---|---|---|---|---|---|---|---|
| bus | 11.1 | 11.9 | 13.0 | **14.0** | 13.1 | 12.6 | 12.0 | 13.3 | 12.5 | 11.9 |
| bicycle | 22.5 | 24.9 | 28.5 | 30.1 | **32.4** | 28.9 | 26.6 | 29.3 | 29.3 | 28.4 |
| emergency-vehicle | 4.5 | 4.6 | 4.3 | **5.4** | 4.5 | 4.4 | 4.1 | 5.2 | 4.0 | 3.9 |
| adult | 34.5 | 43.6 | 56.1 | 58.8 | 59.8 | 46.5 | 36.1 | **60.7** | 56.3 | 49.1 |
| child | 4.2 | **5.1** | 4.6 | 3.5 | 2.8 | 2.6 | 1.9 | 3.6 | 2.9 | 2.7 |
| construction-worker | 13.6 | 16.3 | 22.9 | 25.6 | **28.6** | 24.3 | 20.5 | 27.9 | 25.1 | 22.3 |
| personal-mobility | 6.6 | 9.1 | 8.8 | 8.7 | 9.1 | 6.9 | 6.9 | **10.4** | 8.6 | 8.6 |
| traffic-cone | 44.3 | 46.7 | 50.2 | 52.0 | **54.0** | 51.3 | 48.4 | 53.1 | 52.0 | 50.1 |

Table 4: **Our proposed techniques improve 3D cuboid generation** (Section 4.2). The first row shows results of a method that adopts our finetuned Ground-ingDINO for 2D detection and CM3D (Khurana et al., 2024) for 3D cuboid generation. MHT standards for Multi-Hypothesis Testing for 3D cuboid generation; "SA." uses class-aware sweep aggregation (Table 3); $S_{3D}$ incorporates 3D geometric cues to score generated 3D cuboids; "track" means using 3D tracks to refine scores of generated cuboids. Results demonstrate the effectiveness of each technique in improving 3D cuboid generation.

| MHT | SA. | $S_{3D}$ | track | mAP$^{3D}$ | NDS |
|---|---|---|---|---|---|
| | | | | 18.2 | 23.1 |
| ✓ | | | | 19.2 | 23.8 |
| ✓ | ✓ | | | 20.1 | 24.3 |
| ✓ | ✓ | ✓ | | 20.8 | 24.6 |
| ✓ | ✓ | ✓ | ✓ | **21.9** | **25.0** |

Table 5: **Analysis of learning to refine generated 3D cuboids.** Suppose we manually prepare 3D cuboids on the LiDAR point clouds for the visual examples provided in the annotation guidelines. We use them to learn a model that takes as input a generated 3D cuboid and output refined cuboid. The refinement can translate the cuboid through re-*centering*, adjust cuboid *size*, tune the *orientation*, and re-*score* the cuboid. We tune the model over the validation set. Results show that learning to refine the size of generated cuboids on limited examples is beneficial.

| center | size | orientation | score | mAP$^{3D}$ | NDS |
|---|---|---|---|---|---|
| | | | | 21.9 | 25.0 |
| ✓ | | | | 20.8 | 21.9 |
| | ✓ | | | 21.9 | **26.4** |
| | | ✓ | | 21.9 | 24.7 |
| ✓ | ✓ | | | 21.9 | 25.9 |
| | | | ✓ | 20.3 | 21.2 |

**Assembling the pieces.** We sequentially include the proposed techniques in our pipeline (Fig. 2): the MHT-based 3D cuboid generation, class-aware sweep aggregation, geometry-aided scoring, and tracking-based score refinement. Table 4 shows each technique brings 0.7~1.9 mAP$^{3D}$ gains, and using them all yields 3.7 mAP$^{3D}$ gains. Despite the remarkable improvements, it is also clear that the task AutoExpert is quite challenging that the overall 3D mAP (21.9) is notably lower than supervised learned 3D detectors, e.g., Peri et al. (2023) reports 43.6 mAP on all the 18 classes of nuScenes.

**Does few-shot learning for 3D cuboid refinement work?** To answer this, we assume access to 3D cuboid annotations on LiDAR data corresponding to the visual examples in the annotation guidelines. We train a network on the "few-shot" 3D labeled data, which takes generated 3D cuboids as input and predicts offsets to refine their location, size, orientation and quality score. Despite careful tuning of the network architecture and hyperparameters (details in Section F of the Supplement), the model yields only a small improvement of 1.4 NDS (Table 5). We hypothesize that the scarcity of 3D labeled data poses significant challenges for training a 3D perception model. The results, together with Table 2, suggest that a LiDAR-based FM could substantially advance LiDAR perception.

## 6 CONCLUSION

We introduce *AutoExpert*, a novel and timely benchmark for *Auto-Annotation from Expert-Crafted Annotation Guidelines*. It simulates real-world data annotation pipelines, where human annotators learn from expert-crafted guidelines to label data. We study AutoExpert through 3D LiDAR detection by repurposing the nuScenes dataset. To approach AutoExpert, we present a pipeline and several simple yet effective techniques, including *Multi-Modal Few-Shot Finetuning* and *Multi-Hypothesis Testing-based 3D cuboid generation*, leading to remarkable performance gains over previous approaches. Nevertheless, our results demonstrate that the AutoExpert benchmark remains far from solved, highlighting the need for further research, e.g., developing LiDAR-based foundation models.

## ETHICS STATEMENT

All the authors read the ICLR Code of Ethics and adhere to it. To the best of the authors' knowledge, this work does not have potential violations of the ICLR Code of Ethics.

## REPRODUCIBILITY STATEMENT

For reproducibility, we have released data, code, models and results in this anonymous GitHub repository under the MIT License. Our codebase contains the code of our proposed techniques. The README file clearly lists python command lines used for third-part toolbox setup, data preparation, 2D detector finetuning, 2D mask generation, 3D cuboid generation, and evaluation w.r.t both 2D and 3D metrics. We also provide the intermediate 2D detections and the final 3D detections for fast evaluation without running models.

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

# Appendix

This document supplements the main paper. Below is the outline of this document.

## A   REMARKS, BROADER IMPACTS AND LIMITATIONS

**Remarks on Class Imbalance.** Datasets sourced from the real world often exhibit a class imbalance, containing both common object classes (e.g., `vehicle` and `pedestrian`) and infrequent ones (e.g., `stroller` and `wheelchair`). Consequently, supervised learning methods for 3D detection must be designed to handle this distribution imbalance (Peri et al., 2023). In contrast, AutoExpert itself does not inherently introduce class imbalance, as expert-crafted annotation guidelines provide a similar number of visual examples for each class. However, because foundation models leveraged to solve AutoExpert are pretrained on massive real-world data which follows imbalanced distributions (Parashar et al., 2024), the methods developed for AutoExpert can still inherit data imbalance, leading to biased predictions.

**Remarks on Expert-in-the-loop vs. human-in-the-loop.** AutoExpert does not necessarily mean fully automated annotation without human intervention. Instead, it requires "expert-in-the-loop", e.g., experts design not only the guidelines but also the pipeline for exploiting FMs towards automated data annotation. This is different from human-in-the-loop, which typically means that ordinary human annotators are involved in data annotation. Our work reveals that achieving AutoExpert (or completing highly specialized tasks) requires expert-in-the-loop, i.e., experts must design methods to adapt FMs to improve 2D detection and segmentation, and design geometric constraints.

**Remarks on Datasets and Annotation Guidelines.** Annotation guidelines are rarely made publicly available along with datasets. For instance, neither KITTI (Geiger et al., 2013) nor Argoverse (Wilson et al., 2021) released their annotation guidelines but user guide for challenge competitions; the Waymo Open Dataset (Sun et al., 2020) provides a file called "labeling specifications" which, however, cover only four classes ("vehicle", "pedestrian", "cyclist", and "traffic sign") with only one visual example for each class. These resources serve as user guides for challenge participants to get familiar with their data. It is worth noting that, even the Croissant protocol (Akhtar et al., 2024) has not called dataset contributors to release annotation guidelines. Therefore, we would like to call out for the community to release annotation guidelines in future dataset release.

**Broader Impacts.** Although we commonly believe that pretraining on large-scale data will be the key enabler for generalization to open-world applications, understanding how to appropriately benchmark such methods and pretrained foundation models (FMs) remains challenging. FMs have

been benchmarked in various ways through general tasks such as reasoning, math, open question answering, and physical rule understanding. Our AutoExpert benchmark offers a new venue where various FMs can be accessed in multiple aspects with the final goal of 3D LiDAR detection, e.g., understanding textual descriptions in annotation guidelines, summarizing core information from texts and visual examples, generalizing to specific object classes for precise detection, etc. Our benchmark can facilitate the development of methods for automating data annotation by learning from expert-crafted guidelines. The developed methods can benefit real-world applications which adopt machine learning solutions, where data annotation is typically a prerequisite. Such applications span industry, health care, interdisciplinary research, etc. In the meanwhile, insights, philosophical thoughts and techniques delivered in this work may potentially inspire dataset curation and methods for malicious attacks for specific applications. These could be negative impacts.

**Limitations.** We note several limitations. First, while our pipeline demonstrates promising results, its real-world applicability for data annotation requires further comprehensive validation. Second, our methods do not leverage unlabeled data, which could be exploited through semi-supervised learning to enhance FM adaptation for AutoExpert. Third, we rely on per-category size priors; adapting these priors to individual instances could further improve performance. Fourth, our tracking-based score refinement focuses on cuboid confidence but could also be used for optimizing cuboid orientation or velocity, which are key factors for autonomous driving and evaluation metrics like NDS. Fifth, our 3D cuboid generation operates on isolated object instances, but incorporating contextual and proxemic relationships between objects could yield additional gains. Finally, our work does not attempt to build a LiDAR-based foundation model, a critical yet underexplored direction for future research.

## B  More Details of 2D Detector Finetuning

**Prompt Refinement to Improve Zero-Shot Detection**. We improve prompts towards better zero-shot 2D detection performance with GroundingDINO (Fig. 2). First, we use GPT-4o to generate five synonyms and object size prior for each object class using the prompt template: *"Generate five descriptive terms for objects within the green boxes and provide the average length, width, and height for this category in the real world. The guideline instruction is: [instruction]."*, where *[instruction]* is replaced by actual guideline descriptions. Fig. 7 displays a screenshot of this step. Second, we use each term and their combinations to test GroundingDINO's zero-shot 2D detection performance on the validation set. Third, we select the best term or combination that yields the highest detection precision for each class. Table 6 summarizes the selected terms for each of the 18 nuScenes classes. In our work, we also tested using Qwen (Bai et al., 2023) other than GPT-4o to search for synonyms and object size prior, but we find it produces similar results as GPT-4o. This is likely because certain terms are more frequent in the real world for a given class name that FMs are more familiar with them, so FMs prefer to use these frequent terms to achieve better zero-shot performance (Parashar et al., 2024).

**Few-Shot Finetuning**. With the limited amounts of visual examples available in the annotation guidelines, we finetune GroundingDINO. We test using the original class names and refined names (described the the last paragraph). Refer to the next paragraph for detailed results. We also adopt data augmentation strategies such as random rotation and cropping. Recall that each training image is exclusively annotated with only one class. Hence, when finetuning GroundingDINO, for each training image, we compute the loss only on the focused class and do not count detections of other classes as false positives. We use the validation set for model selection and hyperparameter tuning. The validation set can be thought of as a simulation of expert intervention in real-world annotation scenarios, where experts are overseeing annotation progress and quality, offering timely intervention when needed.

**Detailed Results**. Table 1 in the main paper summarizes comparisons of using different prompts in the off-the-shelf and finetuned GroundingDINO detectors. Here, we provide the full results of these methods in Table 7. The finetuned GroundingDINO (ft-GD) using refined class names (r-name) yields the best results on 12 out of 18 categories w.r.t 3D mAP metric, achieving significant improvements over the zero-shot baseline "GD (o-name)", e.g., on `adult` (from 54.4 to 58.8), `personal-Mobility` (from 0.0 to 9.4), `motorcycle` (from 42.0 to 50.0), and `traffic-cone` (from 44.2 to 52.0).

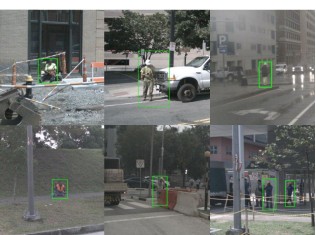

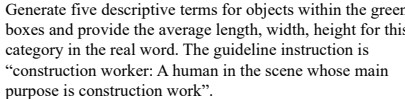

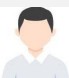

Generate five descriptive terms for objects within the green boxes and provide the average length, width, height for this category in the real word. The guideline instruction is "construction worker: A human in the scene whose main purpose is construction work".

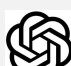

**Five Descriptive Terms for Objects in Green Boxes** :
1. Construction Worker
2. Laborer
3. Builder
4. Hard Hat
5. Mason
**Average Real-World Dimensions**:
- Length : 0.5 meters
- Width: 0.5 meters
- Height : 1.75 meters

Figure 7: **A screenshot of how we search synonyms and object size prior for a given class name.** We use both visual examples and textual description of a specific class available in annotation guidelines. Here, we use the `construction-worker` class as an example.

Table 6: **Refined class names for the nuScenes defined classes**. Notable combinatorial prompts occur in some categories, e.g., the best prompt of `pushable-pullable` is "pushable pullable garbage container" and "hand truck".

| The original class name | The refined class name |
| --- | --- |
| car | car |
| truck | truck |
| trailer | trailer, container |
| bus | bus |
| construction-vehicle | construction-vehicle |
| bicycle | bicycle bike |
| motorcycle | narrow motorcycle |
| emergency-vehicle | police vehicle, emergency vehicle |
| adult | adult |
| child | single little short youth children |
| police-officer | law enforcement officer |
| construction-worker | construction worker, laborer |
| stroller | stroller |
| personal-mobility | personal-mobility, small kick scooter |
| pushable-pullable | pushable pullable garbage container, hand truck |
| debris | debris, full trash bags |
| traffic-cone | traffic_cone |
| barrier | barrier |

## C  SUPERVISED LEARNED 3D DETECTOR ON A DIFFERENT DATASET

In the main paper, we test a model purposefully trained on another dataset, Argoverse2 (AV2) to investigate whether LiDAR-based pretrained models can generalize to specific domains of interest. Concretely, we train a 3D detector CenterPoint (Yin et al., 2021) on the training set of AV2 in a supervised learning way. But this model exhibits severe performance degradation on the nuScenes benchmark (its results in Table 2). Below, we provide implementation details and analysis on its performance.

Table 7: More detailed (per-category) comparisons of results for different strategies of finetuning foundational 2D detectors yield different 3D detection performances. We report the results of zero-shot detector Detic as a reference, which is used in (Khurana et al., 2024).

| class | Detic (o-name) | | GD (o-name) | | ft-GD (o-name) | | GD (r-name) | | ft-GD (r-name) | |
|---|---|---|---|---|---|---|---|---|---|---|
| | mAP$^{2D}$ | mAP$^{3D}$ | mAP$^{2D}$ | mAP$^{3D}$ | mAP$^{2D}$ | mAP$^{3D}$ | mAP$^{2D}$ | mAP$^{3D}$ | mAP$^{2D}$ | mAP$^{3D}$ |
| car | 58.3 | 31.9 | 52.6 | 25.1 | 56.4 | 29.1 | 51.3 | 26.1 | 54.2 | 25.6 |
| truck | 37.2 | 17.6 | 34.3 | 14.2 | 34.4 | 15.0 | 36.3 | 14.1 | 37.5 | 12.6 |
| trailer | 4.2 | 0.8 | 8.5 | 2.0 | 7.0 | 1.3 | 8.1 | 1.8 | 8.5 | 1.7 |
| bus | 59.0 | 6.4 | 59.0 | 5.7 | 60.2 | 8.3 | 59.7 | 5.3 | 59.8 | 6.4 |
| construction-vehicle | 11.1 | 14.7 | 9.9 | 9.7 | 4.5 | 12.0 | 10.2 | 8.9 | 11.0 | 9.5 |
| bicycle | 28.8 | 28.6 | 22.4 | 22.7 | 28.5 | 27.5 | 22.5 | 19.6 | 24.2 | 29.5 |
| motorcycle | 38.1 | 50.9 | 21.7 | 42.0 | 36.4 | 48.1 | 20.8 | 33.5 | 31.5 | 50.0 |
| emergency-vehicle | 0.5 | 0.18 | 1.6 | 0.9 | 4.3 | 2.4 | 11.9 | 1.5 | 14.2 | 2.6 |
| adult | 10.2 | 5.6 | 23.5 | 54.4 | 36.9 | 53.8 | 23.9 | 53.8 | 31.8 | 58.8 |
| child | 0.0 | 0.00 | 0.9 | 0.8 | 1.1 | 0.7 | 6.6 | 3.2 | 5.2 | 3.5 |
| police-officer | 0.0 | 0.1 | 0.0 | 1.7 | 0.3 | 0.7 | 0.3 | 1.0 | 0.8 | 2.2 |
| construction-worker | 0.4 | 2.0 | 5.1 | 27.8 | 9.2 | 28.4 | 4.2 | 28.2 | 5.9 | 25.7 |
| stroller | 1.2 | 7.0 | 13.1 | 27.8 | 13.5 | 20.4 | 14.4 | 29.3 | 15.3 | 21.4 |
| personal-mobility | 0.0 | 0.0 | 0.0 | 0.0 | 0.0 | 0.0 | 0.9 | 0.7 | 10.6 | 9.4 |
| pushable-pullable | 0.0 | 0.0 | 2.6 | 1.2 | 2.9 | 1.1 | 6.1 | 2.8 | 5.8 | 4.8 |
| debris | 0.0 | 0.0 | 0.0 | 0.0 | 0.0 | 0.0 | 0.0 | 0.0 | 0.1 | 0.1 |
| traffic-cone | 51.8 | 50.5 | 46.2 | 44.2 | 52.7 | 40.1 | 46.9 | 43.4 | 52.5 | 52.0 |
| barrier | 0.8 | 0.6 | 3.6 | 9.7 | 2.8 | 8.5 | 2.8 | 8.5 | 5.9 | 11.4 |
| **avg.** | 16.5 | 12.1 | 16.9 | 16.1 | 20.0 | 16.6 | 18.2 | 15.7 | **20.8** | **18.2** |
| **NDS** | 16.6 | | 21.3 | | 21.2 | | 22.1 | | **23.1** | |

Table 8: **Specifics of different LiDAR sensors in nuScenes and Argoverse2 (AV2) for data collection**. Clearly, the LiDAR sensors notably differ in measurement range, horizontal/vertical FOV, point cloud density, and vertical distribution patterns.

| | LiDAR Configuration | |
|---|---|---|
| parameter/feature | nuScenes (Caesar et al., 2020) | AV2 (Wilson et al., 2021) |
| LiDAR Model | Single HDL-32E | Dual VLP-32C |
| Number of channels | 32 | $32 \times 2$ |
| Measurement Range | Max 100m, Effective 70m | Max 200m, Effective 100m |
| Vertical FOV | $-30.67°$ to $+10.67°$ | $-25°$ to $+15°$ |
| Horizontal FOV | $360°$ | $360°$ |
| Scan Frequency | 20Hz | 10Hz |
| Points per Second | $\sim$600k pts/s (20Hz $\times$ 30k/frame) | $\sim$1.2M pts/s (10Hz $\times$ 120k/frame $\times$ 2) |
| Vertical Resolution | $0.4°$ (center), $0.4°$ to $2.08°$ (edge) | $0.33°$ (center), $1.0°$ (edge) |
| Horizontal Resolution | $0.32°$ (20Hz) | $0.096°$ (10Hz) |
| Intensity Range | 0-255 | 0-255 |

**CenterPoint Training on the Argoverse2 Dataset**. The Argoverse2 (AV2) dataset annotates LiDAR data with 3D cuboids for 30 object classes. To train a CenterPoint model that can be applied to nuScenes data, we unify the data format and class vocabulary of AV2 according to nuScenes. Then, we supervised-learn the 3D LiDAR detector CenterPoint on the training set of AV2. After training, we apply it to the AutoExpert test-set. Table 2 reports its results, showing that CenterPoint significantly underperforms our method and existing approaches.

**Analysis**. We analyze the LiDAR models of the AV2 and nuScenes, finding that they have different LiDAR sensor parameters (Table 8). As a result, the LiDAR sensors capture data that is different in (1) measurement range, (2) horizontal/vertical FOV, (3) point cloud density, (4) intensity sensitivity, (5) acquisition frequency. All these make AV2 LiDAR data different in distribution from the nuScenes LiDAR data, explaining the poor performance of AV2-trained CenterPoint. This further demonstrates the challenge and the need of training LiDAR foundation models.

## D  DETAILS AND ANALYSES OF MHT-BASED 3D CUBOID GENERATION

We provide more details of the proposed 3D cuboid generation method based on Multiple Hypotheses Testing (MHT). First, for each 2D detection (denoted by $B_{2D}$) of GroundingDINO, we determine a cuboid dimension $(l_0, w_0, h_0)$ which is obtained by using GPT-4o (Fig. 7). We also use SAM to

Table 9: Sensitivity analysis of rotation and translation step size parameters. Based on bolded values, we set the corresponding translation and rotation step sizes as default, which provide good trade-off between computational cost and detection performance.

| Rotation Step Size (in radian) | | | | | |
| --- | --- | --- | --- | --- | --- |
| Rotation Step | $\pi/40$ | $\pi/30$ | $\pi/20$ | **$\pi/10$** | $\pi/5$ |
| mAP$^{3D}$ | 19.3 | 19.3 | 19.2 | **19.2** | 18.7 |
| NDS | 24.0 | 23.9 | 23.8 | **23.8** | 23.2 |
| Translation Step Size (in meter) | | | | | |
| Translation Step | 0.3m | **0.5m** | 0.8m | 1.0m | 1.5m |
| mAP$^{3D}$ | 19.4 | **19.2** | 18.6 | 18.3 | 16.4 |
| NDS | 23.8 | **23.8** | 23.3 | 22.9 | 20.9 |

obtain the object mask. Further, we determine a frustum based on the 2D detection and camera and LiDAR parameteres. Then, within this frustum, we run DBSCAN (Schubert et al., 2017) to cluster LiDAR points (denoted in set $P$) that are projected onto the foreground mask in the image plane. Third, we find the largest cluster and place the 3D cuboid with its center being the cluster of the cluster centroid. Fourth, we carry out MHT detailed below.

We predefine stepsizes w.r.t translation and rotation (analyzed in the next paragraph). Hence, during MHT, we vary parameters $\theta = [x, y, z, \psi]$ (cuboid center location and cuboid orientation) and compute the point ratio of points falling in the cuboid $B_{3D}(\theta)$:

$$R(\theta) = \frac{1}{|P|} \sum_{p_i \in P} \mathbb{I}(p_i \in B_{3D}(\theta)), \tag{1}$$

where $\mathbb{I}(\cdot)$ is the indicator function. Moreover, we also the intersection-over-union (IoU) between the projected 3D cuboid on the image plane (denoted as $\pi(B_{3D}(\theta))$ and the 2D box $B_{2D}$ output by GroundingDINO: $\text{IoU}_\theta(\pi(B_{3D}(\theta)), B_{2D})$. Lastly, we select a cuboid that maximize the above two metrics with parameter $\theta^*$:

$$\theta^* = \arg\max_\theta R(\theta) + \text{IoU}_\theta(\pi(B_{3D}(\theta)), B_{2D}) \tag{2}$$

**Sensitivity analysis**. We conduct a comprehensive sensitivity analysis of the rotation and translation step size parameters in the proposed MHT-based approach. The analysis aims to determine the optimal parameter values that balance computational efficiency and 3D detection performance.

Table 9 demonstrates that our method is robust to a large range of parameter variations. For rotation step size, values between $\pi/40$ and $\pi/10$ radians yield good and similar performance, with $\pi/10$ radians selected as the default value owing to its favorable balance between computational efficiency and detection accuracy. Similarly, for translation step size, values between 0.3m and 0.8m maintain good and stable performance, with 0.5m chosen as the default value in our experiments. Moreover, the sensitivity analysis confirms that coarse step sizes ($\pi/5$ radians for rotation or 1.0m+ for translation) lead to significant performance degradation, while excessively fine step sizes offer diminishing returns and substantially increased computational costs.

## E  3D Detection Methods and Their Detailed Results

**Methods**. SAM3D (Zhang et al., 2023a) utilizes Bird's-Eye-View (BEV) images of LiDAR sweeps as input to the pre-trained SAM model (Kirillov et al., 2023) for segmenting salient objects. When applied to the nuScenes dataset, it successfully segments sufficiently large objects such as `vehicle` but fails to detect small classes such as `child` and `traffic-cone`. Oyster (Zhang et al., 2023b) and CPD (Wu et al., 2024b) are unsupervised learning methods that were developed and benchmarked on the Waymo dataset (which contains only three classes: vehicle, pedestrian and cyclist). They struggle to detect diverse object classes. LISO (Baur et al., 2024) is a self-supervised LiDAR 3D detector on movable objects without class label prediction. Both LISO and CPD generate pseudo-labels based on geometric priors (e.g., class shape and size) and use these pseudo-labels to train their

Table 10: Comparison of 3D detection / cuboid generation methods w.r.t diverse metrics. Following nuScenes, we report results w.r.t metrics including mean Average Precision (mAP$^{3D}$) and the nuScenes Detection Score (NDS), along with detailed error metrics: mean Translation Error (mATE), mean Scale Error (mASE), mean Orientation Error (mAOE), mean Velocity Error (mAVE), and mean Attribute Error (mAAE). The results are comparable to those in Table 1, 2, and 4. In addition to compared methods, we also list our developed models in this work (refer to captions of Table 1 and 4 for these models).

| Method | mATE ↓ | mASE ↓ | mAOE ↓ | mAVE ↓ | mAAE ↓ | mAP$^{3D}$ ↑ | NDS ↑ |
|---|---|---|---|---|---|---|---|
| SAM3D (Zhang et al., 2023a) | 0.812 | 0.762 | 1.512 | 1.312 | 0.812 | 0.016 | 0.069 |
| Oyster (Zhang et al., 2023b) | 0.755 | 0.715 | 1.451 | 1.201 | 0.771 | 0.063 | 0.107 |
| Oyster w/ frustum | 0.723 | 0.651 | 1.563 | 1.008 | 0.711 | 0.129 | 0.156 |
| LISO (Baur et al., 2024) | 0.725 | 0.679 | 1.411 | 1.152 | 0.733 | 0.089 | 0.131 |
| LISO w/ frustum | 0.679 | 0.583 | 1.429 | 0.903 | 0.644 | 0.157 | 0.198 |
| CPD (Wu et al., 2024b) | 0.718 | 0.658 | 1.408 | 1.108 | 0.708 | 0.101 | 0.142 |
| CPD w/ frustum | 0.641 | 0.549 | 1.338 | 0.886 | 0.590 | 0.179 | 0.223 |
| Centerpoint (Yin et al., 2021) | 0.971 | 0.517 | 0.794 | 0.546 | 0.447 | 0.036 | 0.190 |
| CM3D (w/ Detic) (Khurana et al., 2024) | 0.775 | 0.587 | 1.189 | 1.084 | 0.579 | 0.121 | 0.166 |
| GD w/ o-name + CM3D | 0.669 | 0.568 | 1.368 | 0.869 | 0.566 | 0.161 | 0.213 |
| ft-GD w/ o-name + CM3D | 0.657 | 0.564 | 1.337 | 0.942 | 0.550 | 0.166 | 0.212 |
| GD w/ r-name + CM3D | 0.639 | 0.548 | 1.349 | 0.833 | 0.551 | 0.157 | 0.221 |
| ft-GD w/ r-name + CM3D | 0.636 | 0.543 | 1.322 | 0.875 | 0.548 | 0.182 | 0.231 |
| ft-GD w/ r-name + MHT | 0.570 | 0.538 | 1.130 | 0.920 | 0.555 | 0.192 | 0.238 |
| ft-GD w/ r-name + MHT + SA | 0.567 | 0.538 | 1.155 | 0.924 | 0.549 | 0.201 | 0.243 |
| ft-GD w/ r-name + MHT + SA + $S_{3D}$ | 0.559 | 0.540 | 1.150 | 0.928 | 0.555 | 0.208 | 0.246 |
| ft-GD w/ r-name + MHT + SA + $S_{3D}$ + track | **0.565** | **0.537** | **1.124** | **0.943** | **0.554** | **0.219** | **0.250** |

Table 11: Per-category results (mAP$^{3D}$) of different 3D cuboid generation methods. We list not only our MHT-based method but also our final method. Our methods resoundingly outperform the compared approaches.

| Class | SAM3D | Oyster | Oyster w/ frustum | LISO | LISO w/ frustum | CPD | CPD w/ frustum | Centerpoint | CM3D | MHT | Our Final |
|---|---|---|---|---|---|---|---|---|---|---|---|
| car | 6.2 | 13.1 | 20.1 | 19.0 | 25.0 | 20.7 | 26.5 | 15.6 | 25.6 | 30.0 | **31.8** |
| truck | 5.2 | 4.1 | 9.0 | 7.0 | 12.8 | 7.7 | 13.9 | 4.1 | 12.6 | 15.4 | **16.7** |
| trailer | 0.2 | 0.4 | 1.2 | 0.8 | 1.4 | 0.9 | 1.6 | 0.2 | 1.7 | 1.7 | **1.9** |
| bus | 2.1 | 2.7 | 4.6 | 6.4 | 11.7 | 7.2 | 12.9 | 2.2 | 6.4 | 14.0 | **15.5** |
| construction-vehicle | 0.5 | 3.1 | 6.8 | 4.3 | 7.9 | 4.6 | 8.3 | 1.0 | 9.5 | 9.5 | **9.9** |
| bicycle | 3.3 | 10.0 | 21.1 | 13.7 | 25.1 | 14.4 | 27.8 | 2.0 | 29.5 | 30.1 | **33.4** |
| motorcycle | 4.1 | 17.1 | 35.7 | 23.1 | 42.3 | 22.6 | 42.7 | 8.4 | 50.0 | 50.7 | **51.2** |
| emergency-vehicle | 0.1 | 0.8 | 1.9 | 2.5 | 4.5 | 2.5 | 4.5 | 0.2 | 2.6 | 5.4 | **5.4** |
| adult | 0.9 | 19.8 | 42.0 | 26.7 | 47.2 | 27.9 | 52.3 | 15.8 | 58.8 | 58.8 | **62.7** |
| child | 0.1 | 1.2 | 2.5 | 1.6 | 2.9 | 2.5 | 4.5 | 0.0 | 3.5 | 3.5 | **5.4** |
| police-officer | 0.2 | 0.9 | 1.6 | 1.0 | 1.8 | 1.6 | 3.0 | 2.0 | 2.2 | 2.2 | **3.6** |
| construction-worker | 0.2 | 8.6 | 18.4 | 11.7 | 21.4 | 13.3 | 25.9 | 3.6 | 25.7 | 25.7 | **31.1** |
| stroller | 1.5 | 6.9 | 15.3 | 9.8 | 17.9 | 13.1 | 27.3 | 0.4 | 21.4 | 21.5 | **32.7** |
| personal-mobility | 0.9 | 3.7 | 6.7 | 4.0 | 7.3 | 5.9 | 10.7 | 0.0 | 9.4 | 8.7 | **12.8** |
| pushable-pullable | 0.1 | 0.1 | 0.1 | 0.1 | 0.1 | 0.1 | 4.3 | 0.0 | 0.1 | 0.1 | **5.2** |
| debris | 0.1 | 0.1 | 0.1 | 0.1 | 0.1 | 0.1 | 0.1 | 0.0 | 0.1 | 0.1 | **0.1** |
| traffic-cone | 1.6 | 16.9 | 37.1 | 23.6 | 43.3 | 27.6 | 47.7 | 8.7 | 52.0 | 52.0 | **57.2** |
| barrier | 1.5 | 3.9 | 8.1 | 5.2 | 9.6 | 9.0 | 14.1 | 3.7 | 11.4 | 11.5 | **16.9** |
| **mAP$^{3D}$** | 1.6 | 6.3 | 12.9 | 8.9 | 15.7 | 10.1 | 17.9 | 3.6 | 18.2 | 19.2 | **21.9** |
| **NDS** | 6.9 | 10.7 | 15.6 | 13.1 | 19.8 | 14.2 | 22.3 | 19.0 | 23.1 | 23.8 | **25.0** |

detectors. Note that LISO, Oyster and CPD all adopt DBSCAN clustering for 3D box fitting (Zhang et al., 2017; You et al., 2022). The 3D detector CenterPoint (Yin et al., 2021) is trained on the Argoverse2 training set dataset in a supervised way (details in Appendix C).

To adapt the above methods to AutoExpert data for detecting objects of 18 classes, we make necessary modifications for them. As some of these methods do not predict class labels for detected 3D cuboids, we use the finetuned GroundingDINO to assign labels. Specifically, for each generated 3D cuboid, we project it onto the 2D image plane, identify a matched 2D detection by GroundingDINO, and assign the corresponding class label to this 3D cuboid. Moreover, LISO, Oyster and CPD produce 3D cuboids directly on LiDAR sweeps; we improve them by exploiting GroundingDINO's 2D detections to limit searching space in LiDAR points. Concretely, for each 2D detection, we determine the frustum and run each of these methods to generate a 3D cuboid. We mark the modified method with " w/ frustum". This modification fairly compares our MHT-based 3D cuboid generation method with

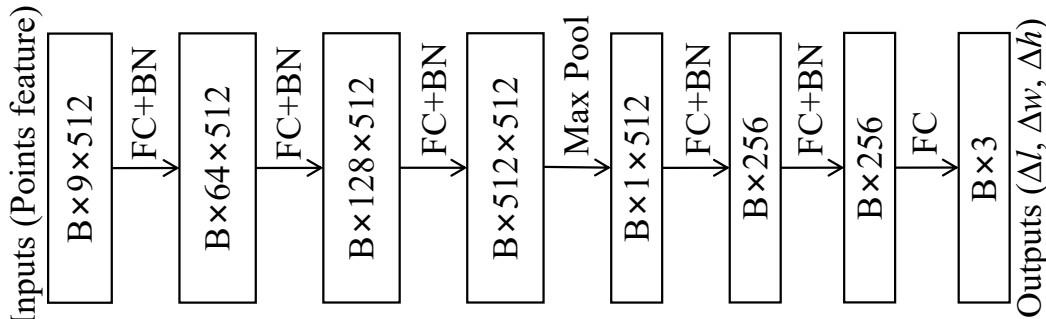

Figure 8: **Architecture of the proposed PointNet (Qi et al., 2017) model $P_\phi$ for 3D cuboid refinement**. Here, B denotes the batch size, FC corresponds to a fully connected layer, and BN represents a batch normalization layer. The input to the model consists of 512 LiDAR points, each represented by a 9-dimensional feature vector (as defined in Equation 4). If the number of points in the frustum point cloud is fewer than 512, random oversampling is applied to reach the target count of 512 points. Conversely, if the point cloud contains more than 512 points, random downsampling is performed to reduce the number to 512. The model outputs dimensional offsets used to refine the size produced by our MHT-based 3D cuboid generation method.

these existing ones, as all the methods exploit GroundingDINO and only differ in the 3D cuboid generation algorithm. Table 10 compares these methods with out MHT-based methods w.r.t the NDS metric and more nuanced metrics. Table 11 lists the per-class 3D detection performance of these methods. The results demonstrate that our MHT-based 3D cuboid generation method resoundingly outperforms all compared approaches.

## F FEW-SHOT SUPERVISED LEARNING OF A 3D REFINEMENT MODEL

Assuming access to 3D annotated LiDAR data for the few-shot visual images available in annotation guidelines, we study whether training a lightweight model on such data can refine 3D detections. We design a PointNet network $P_\phi$ (Qi et al., 2017), which takes as input the 3D locations of LiDAR points and outputs 3D cuboid dimension's offset (and optionally orientation offset, confidence score, and cuboid center offset). Fig. 8 illustrates the architecture of this PointNet model.

To prepare such training data, we apply our finetuned GroundingDINO, SAM, and the MHT-based 3D cuboid generation method to the limited amount of training images. Hence, for each 2D detection, we obtain its corresponding object mask, frustum, and 3D cuboid dimension $\mathbf{d}_0 = [l_0; w_0; h_0]$, its center location $[x_0; y_0; z_0]$ and orientation $\theta_0$. For this 2D detection, we have the input data to the PointNet as the set of 3D locations of LiDAR points from the corresponding frustum. Importantly, we transform these 3D locations (say $\mathbf{p}_{\text{trans}} = [x; y; z]$) depending on the 3D cuboid center location and orientation:

$$\mathbf{p}_{\text{trans}} = \begin{bmatrix} \cos\theta_0 & \sin\theta_0 & 0 \\ -\sin\theta_0 & \cos\theta_0 & 0 \\ 0 & 0 & 1 \end{bmatrix} \left( \begin{bmatrix} x \\ y \\ z \end{bmatrix} - \begin{bmatrix} x_0 \\ y_0 \\ z_0 \end{bmatrix} \right). \tag{3}$$

With the transformed coordinates of LiDAR points, for each point, we construct a 9-dim feature $\mathbf{f}$ using below

$$\mathbf{f} = \begin{bmatrix} \mathbf{p}_{\text{trans}}; \mathbf{d}_0 - \mathbf{p}_{\text{trans}}; \mathbf{d}_0 + \mathbf{p}_{\text{trans}} \end{bmatrix} \tag{4}$$

The network $P_\phi$ outputs offset $\Delta d$ between the ground-truth dimension $l_{\text{gt}}, w_{\text{gt}}, h_{\text{gt}}$ and the dimension of the initial 3D cuboid in log scale:

$$\Delta l = \log\left(\frac{l_{\text{gt}}}{l_0}\right), \quad \Delta w = \log\left(\frac{w_{\text{gt}}}{w_0}\right), \quad \Delta h = \log\left(\frac{h_{\text{gt}}}{h_0}\right), \tag{5}$$

When training the network, we adopt a smooth L1 loss with the default $beta$ hyperparameter 1.0.

Table 12: Comparative analysis of occlusion robustness (mAR$^{3D}$) and distance performance (mAP$^{3D}$). Our MHT method consistently outperforms CM3D, with particularly significant gains for heavily occluded objects (60-100% visibility) and distant targets (20-30m). The final method incorporating LiDAR aggregation and tracking-based refinement demonstrates substantial improvements, especially for far-field and occluded objects.

| Method | mAR$^{3D}$/Occlusion Level | | | | mAP$^{3D}$/Distance (m) | | | |
|---|---|---|---|---|---|---|---|---|
| | 60-100% | 40-60% | 20-40% | 0-20% | 0-10 | 10-20 | 20-30 | 0-50 |
| CM3D (Khurana et al., 2024) | 33.5% | 49.4% | 57.2% | 57.6% | 24.2 | 22.1 | 12.8 | 18.2 |
| MHT (Ours) | 34.1% (+0.6%) | 51.5% (+2.1%) | 58.1% (+0.9%) | 60.9% (+3.3%) | 26.5 (+2.3) | 23.0 (+0.9) | 13.9 (+1.1) | 19.2 (+1.0) |
| Our Final Method | **36.5%** (+3.0%) | **54.0%** (+4.6%) | **59.9%** (+2.7%) | **61.9%** (+4.3%) | **26.2** (+2.0) | **25.7** (+3.6) | **17.1** (+4.3) | **21.9** (+3.7) |

## G   ANALYSIS OF OUR MHT-BASED 3D CUBOID GENERATION FOR OCCLUDED AND FAR-FIELD OBJECTS

We analyze the performance of our MHT-based 3D cuboid generation method for occluded and far-field objects. We provide quantitative evaluations using nuScenes' visibility tags and distance stratification on the AutoExpert test set. As our method and previous approaches such as CM3D do not predict occlusion levels, computing Average Precision (AP) for occlusion analysis is not appropriate. Instead, we report mean Average Recall (mAR$^{3D}$), following the nuScenes protocol to average over distance thresholds {0.5, 1.0, 2.0, 4.0} meters across all 18 classes. For distance analysis, we report mAP$^{3D}$ averaged over these thresholds.

Tabel 12 provides breakdown results of our method and the compared CM3D. Our method consistently outperforms CM3D across all occlusion levels and distance ranges. Importantly, our final method, which incorporates LiDAR aggregation and tracking-based refinement, yields particularly large performance gains for heavily occluded objects (60-100% visibility) and distant targets (20-30m). These improvements can be attributed to the densification of LiDAR points through aggregation and refinement techniques, which significantly aid in detecting far-field small objects and occluded targets. This analysis validates the robustness of our approach in challenging scenarios involving occlusion and long distance.

## H   COMPUTATIONAL EFFICIENCY ANALYSIS

One may think our MHT-based 3D cuboid generation method is computationally expensive as it tests multiple cuboid candidates in order to select the most favorable one. In fact, this method is quite efficient with our optimized implementation — we have optimized the MHT implementation using the Numba compiler with GPU parallelization for calculating point coverage, achieving significant speed improvements. The implementation details are available in our released code. Next, we conduct the analysis on a single NVIDIA A6000 GPU by comparing against the recent work CM3D. We report the averaged time cost over sweeps of AutoExpert test set.

Table 13 lists the wall-clock time of our method and the compared CM3D, both of which exploit the 2D detector GroundingDIMO, SAM for foreground segmentation, and their respective 3D cuboid generation. Clearly, the 2D detection stage constitutes the majority of computation time (1.42 seconds, 34.6% of total), indicating the primary computational bottleneck for both methods. Moreover, our MHT implementation adds approximately 1.02 seconds overhead in 3D cuboid generation compared to CM3D, increasing from 1.59 seconds to 2.61 seconds. Despite this increase, the total processing time of 4.10 seconds per LiDAR sweep remains practical from the perspective of automated data annotation, which prioritizes accuracy rather than inference speed.

It is worth noting that further optimizations are possible, e.g., (1) replacing the current 2D detector with more efficient alternatives to reduce the 2D detection time, (2) streamlining the processing pipeline through batched operations to improve overall throughput.

Table 13: Wall-clock time comparison for processing a single LiDAR sweep. The 2D detection stage dominates the computation time, while our MHT approach adds reasonable overhead in the 3D cuboid generation phase.

| Method | 2D Detection (seconds) | Segmentation (seconds) | 3D Cuboid Generation (seconds) | Total Time (seconds) |
|---|---|---|---|---|
| CM3D (Khurana et al., 2024) | 1.42 | 0.072 | 1.59 | 3.08 |
| MHT (Ours) | 1.42 | 0.072 | 2.61 | 4.10 |

Table 14: **Analysis of sweep aggregation strategies on per-class 3D detection performance** (mAP$^{3D}$). "$P$+C+$F$" denotes aggregating the past $P$ sweeps, the current sweep $C$, and the future $N$ sweeps; we drop $P$ or $F$ if not aggregating any past or future sweeps. In each row, we bold the highest number and highlight it if exceeding other numbers by 0.5 points. Somewhat surprisingly, aggregation strategies greatly impacts performance on certain classes, e.g., `construction-worker`, `bicycle` and `traffic-cone`, aggregating the past 2 sweeps yields remarkably better performance than other strategies.

| Class | 10+C | 6+C | 2+C | C | C+2 | C+6 | C+10 | 1+C+1 | 3+C+3 | 5+C+5 |
|---|---|---|---|---|---|---|---|---|---|---|
| car | 25.6 | 27.4 | 29.1 | **30.0** | 29.8 | 28.6 | 27.1 | 29.9 | 28.8 | 28.1 |
| truck | 14.0 | 14.6 | **15.5** | 15.4 | 15.4 | 15.0 | 14.3 | **15.5** | 15.3 | 14.9 |
| trailer | 1.7 | 1.7 | 1.7 | 1.7 | 1.7 | 1.7 | 1.7 | 1.7 | 1.7 | 1.7 |
| bus | 11.1 | 11.9 | 13.0 | **14.0** | 13.1 | 12.6 | 12.0 | 13.3 | 12.5 | 11.9 |
| construction-vehicle | 8.8 | 9.2 | 9.5 | 9.5 | 9.5 | 9.4 | 9.1 | 9.5 | 9.6 | 9.4 |
| bicycle | 22.5 | 24.9 | 28.5 | 30.1 | **32.4** | 28.9 | 26.6 | 29.3 | 29.3 | 28.4 |
| motorcycle | 37.6 | 42.2 | 48.5 | 50.7 | 49.6 | 43.6 | 38.0 | **51.2** | 48.6 | 45.5 |
| emergency-vehicle | 4.5 | 4.6 | 4.3 | **5.4** | 4.5 | 4.4 | 4.1 | 5.2 | 4.0 | 3.9 |
| adult | 34.5 | 43.6 | 56.1 | 58.8 | 59.8 | 46.5 | 36.1 | **60.7** | 56.3 | 49.1 |
| child | 4.2 | **5.1** | 4.6 | 3.5 | 2.8 | 2.6 | 1.9 | 3.6 | 2.9 | 2.7 |
| police-officer | 1.2 | 1.5 | 2.2 | 2.2 | 2.2 | 2.0 | 1.8 | **2.3** | 2.0 | 1.9 |
| construction-worker | 13.6 | 16.3 | 22.9 | 25.6 | **28.6** | 24.3 | 20.5 | 27.9 | 25.1 | 22.3 |
| stroller | 19.2 | 20.2 | 21.5 | 21.5 | 23.3 | 24.1 | **24.2** | 21.7 | 21.5 | 23.2 |
| personal-mobility | 6.6 | 9.1 | 8.8 | 8.7 | 9.1 | 6.9 | 6.9 | **10.4** | 8.6 | 8.6 |
| pushable-pullable | 4.6 | 4.7 | 4.7 | 4.8 | **5.0** | 4.8 | 4.8 | 4.8 | 4.8 | 4.8 |
| debirs | 0.1 | 0.1 | 0.1 | 0.1 | 0.1 | 0.1 | 0.1 | 0.1 | 0.1 | 0.1 |
| traffic-cone | 44.3 | 46.7 | 50.2 | 52.0 | **54.0** | 51.3 | 48.4 | 53.1 | 52.0 | 50.1 |
| barrier | 11.3 | 11.4 | 11.5 | 11.5 | 11.5 | 11.5 | 11.5 | 11.5 | 11.5 | 11.5 |

## I    FULL RESULTS OF DIFFERENT LiDAR SWEEP AGGREGATION STRATEGIES

Table 14 provides with per-clas mAP$^{3D}$ by different LiDAR sweep aggregation strategies, supplementing Table 3 in the main paper.

## J    MORE VISUALIZATIONS

Fig. 9 visualizes more visual results on the nuScenes dataset (Caesar et al., 2020).

## K    IMAGE EXAMPLES IN GUIDELINES

Fig. 10 illustrates example images of 5 categories from the annotation guidelines (3 examples visualized per category). The complete set of image examples are provided in the supplementary material. These image examples serve as training data for finetuning the foundational 2D detector.

## L    OPEN-SOURCE CODE AND MORE EXPERIMENTAL DETAILS

**Open-Source Code**. We include our codebase as a part of the supplementary material, refer to the `README.md` file for instructions of running the code. We also include the training images (i.e., those included in annotation guidelines). We do not include model weights in the supplementary material as they exceed the space limit (100MB). We have made an anonymous GitHub repository to host our open-source code, data, results, and models under the MIT License.

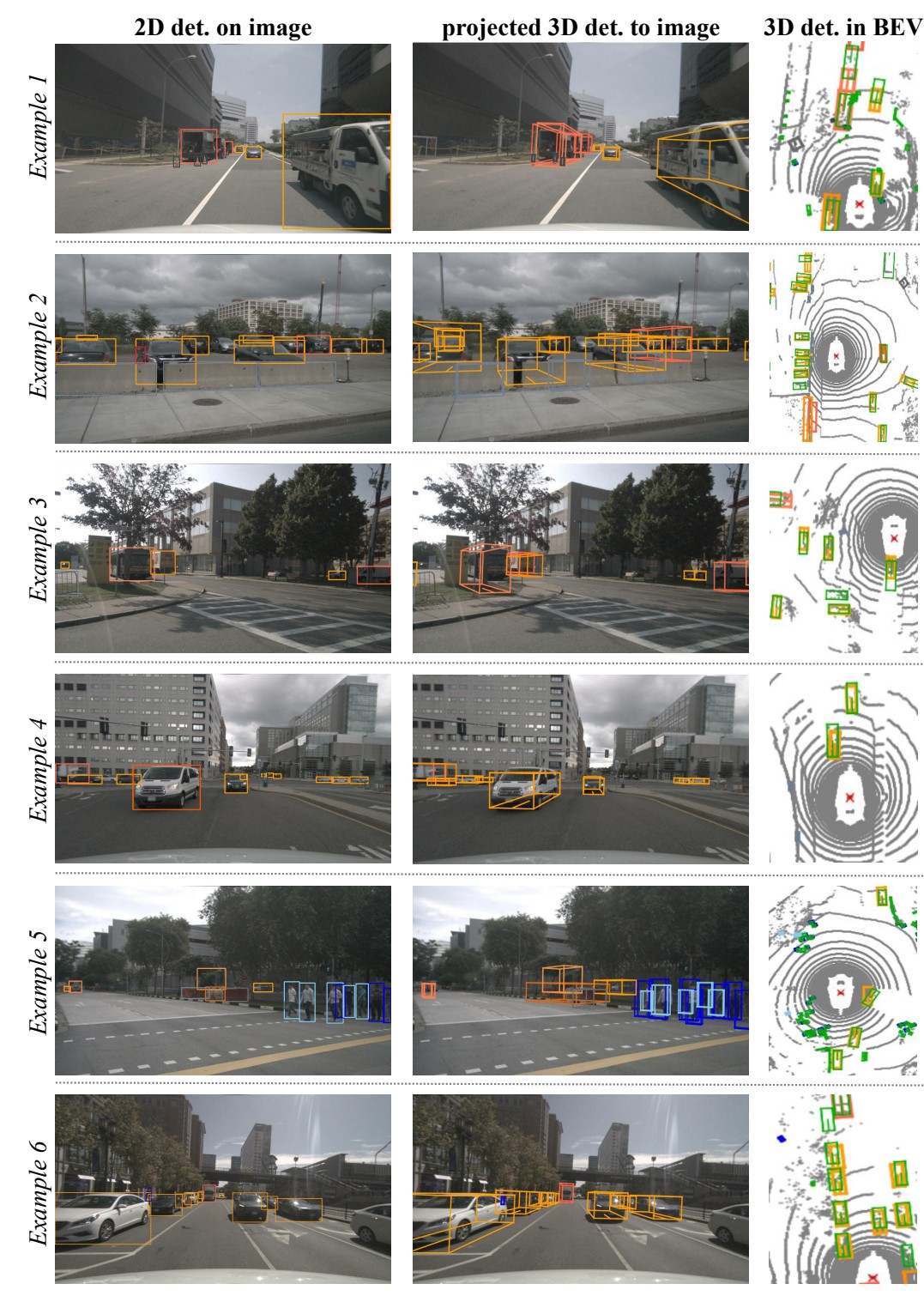

| 2D det. on image | projected 3D det. to image | 3D det. in BEV |

*Example 1*
*Example 2*
*Example 3*
*Example 4*
*Example 5*
*Example 6*

Figure 9: More visual results of 2D detection and generated 3D cuboids (i.e., 3D detection) using our method.

**More Experimental Details**. The training environment uses Python 3.8.20 with PyTorch 2.4.1+cu121 and 4 compute workers. We employ an AdamW optimizer (lr=0.0001, weight_decay=0.0001) on a

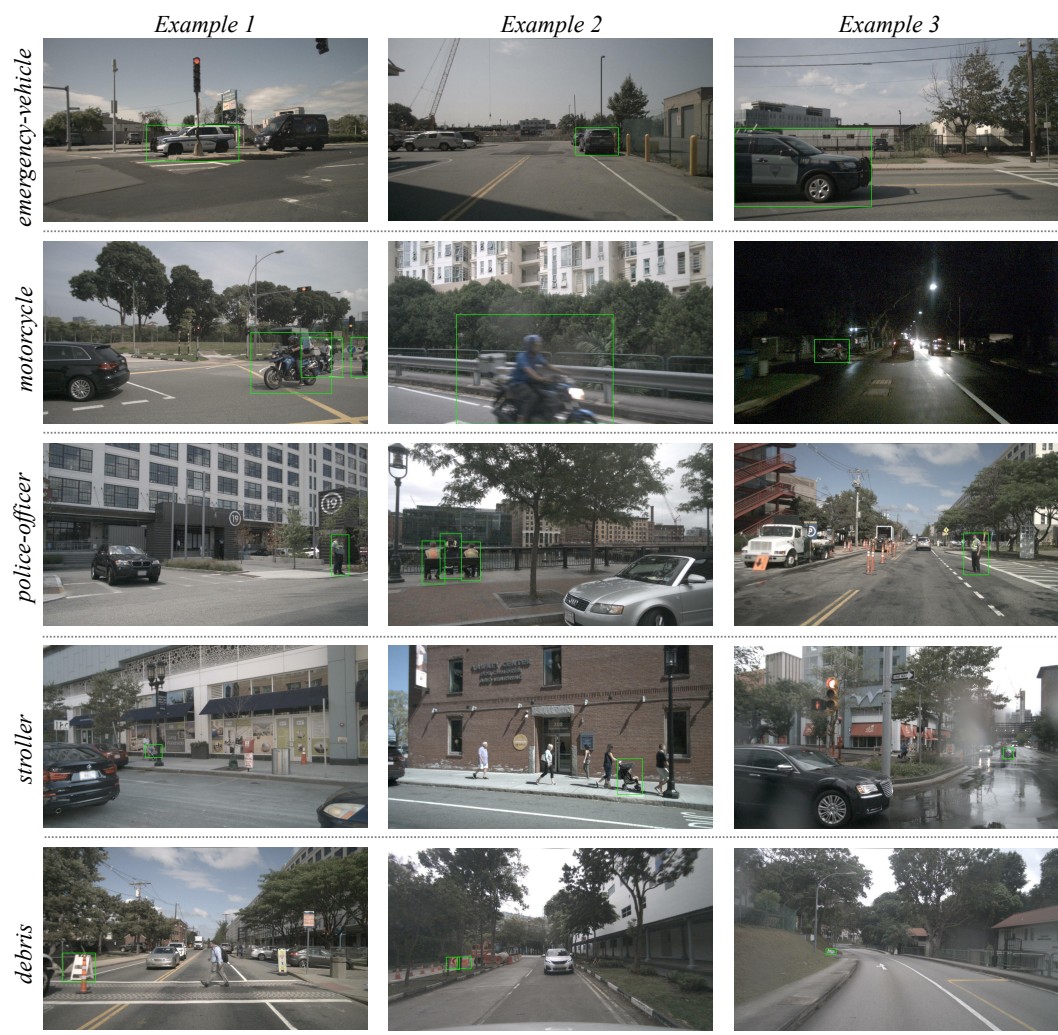

Figure 10: **Images examples in annotation guidelines**. We present 5 out of the 18 nuScenes categories, with 3 example images shown for each category, where the green bounding boxes are 2D annotations for objects of the corresponding classes. It is worth noting the federated annotation: taking the emergency-vehicle category as an example, even if objects of the car category appear in the images, no corresponding annotations are provided.

single NVIDIA A6000 GPU with 40GB memory, reaching peak GPU memory usage of 26GB when training GroundingDINO. The dataset of training GroundingDINO comprises 112 training images (∼ 6 per class), 570 validation images (∼ 32 per class), and 36,114 test images (6,019 LiDAR frames × 6 images). End-to-end training GroundingDINO with online validation takes approximately 2 hours.

