# OpenReview forum: "Auto-Annotation from Expert-Crafted Guidelines: A Benchmark through 3D LiDAR Detection"
_ICLR.cc/2026/Conference — ICLR 2026 Conference Withdrawn Submission_

### Official Review · Reviewer_NRNV · 2025-10-16

**Soundness:** 3
**Presentation:** 2
**Contribution:** 3
**Rating:** 4
**Confidence:** 4

**Summary:**

The paper proposes AutoExpert, a benchmark for auto-annotation from expert-crafted guidelines in 3D LiDAR detection. It repurposes the nuScenes dataset and its official annotation guidelines (18 classes), noting that the guidelines include only textual descriptions and a few visual examples, without 3D bounding boxes, thereby framing a multimodal few-shot setting for 3D detection

The authors build a pipeline that (i) uses GPT-4o to derive class synonyms and per-class size priors, (ii) adapts GroundingDINO for 2D detection and SAM for instance masks, and (iii) lifts 2D to 3D via a Multi-Hypothesis Testing (MHT) cuboid search augmented with sweep aggregation, geometry-aided scoring, and tracking-based score refinement.

**Strengths:**

1. Timely benchmark capturing a realistic labeling workflow, including a few visual exemplars + textual policy, no 3D cuboids in guidelines; 18 classes from the nuScenes official guide.
2. Simple, composable pipeline that practitioners can reproduce: VLM-assisted prompt/size priors, GD-based 2D, SAM masks, MHT in 3D with sweep aggregation and tracking; each piece shows incremental gains (final mAP 21.9/NDS 25.0).
3. Diagnostic studies on sweep aggregation asymmetries by class (Table 3/14), occlusion/distance analysis, and compute breakdown (4.10 s per sweep; 2D stage dominates)

**Weaknesses:**

1. Potential split leakage/evaluation ambiguity (major). The paper samples 570 frames from the official validation set for hyper-parameter tuning and then uses the official validation set as a test. Unless the 570 frames are excluded, this biases the results. Authors must clarify and release explicit split IDs.
2. The evaluation scope is narrow. All main results are on nuScenes; the “test” is the official validation set (not a held-out test), which could inflate optimism under extensive model selection.
3. Limited novelty in core algorithm. The 3D lifting (frustum + size prior + search) and DBSCAN usage are established ideas; the contribution is primarily one of integration.
4. Dependence on closed-source LLMs (GPT-4o, etc.) without a robustness study. Size priors (e.g., police officer 0.5×0.5×1.75 m) come from GPT-4o; failure modes if priors are biased or off-distribution are not probed.
5. Absolute accuracy remains low. Despite careful engineering, the final mAP3D 21.9 remains far below supervised reference (43.6), limiting near-term utility for precise labeling without substantial expert QA.
6. Typos and polish. Multiple typographical errors harm presentation quality (non-exhaustive):
"we" in the abstract should be the capital We. “occlusders” → occluders (Fig. 4 caption). “police-offier” in Fig. 3 panel text. “Tabel 12” → Table 12. “GroundingDIMO” → GroundingDINO (Table 13 caption text).  “per-clas mAP3D” → per-class mAP3D, and “debirs” → debris in Table 14 row label.
7. Heuristic-heavy 3D lifting. MHT uses discrete rotation/translation grids and hand-crafted scoring/tracking; while ablations show robustness to step sizes, this remains engineering-driven, with limited learning beyond 2D GD finetuning.
8. Compute overhead. The full method raises end-to-end time from 3.08 s to 4.10 s per sweep versus CM3D (major bottleneck in 2D detection, but MHT adds ~1.02 s). For large-scale annotation, this cost merits discussion.

**Questions:**

1. How sensitive is MHT to wrong priors? Report performance when perturbing LWH by ±20–40% per class.
2. Do the 570 validation frames overlap with the “test” set? Please provide list of sample IDs and confirm no leakage.
3. You mention Qwen yields similar prompts—can you reproduce Table 1/2 with Qwen-only to ensure reproducibility without GPT-4o?
4. For practical labeling, what fraction of cuboids passes expert QA without edits? Any per-class QA cost savings?

---

### Official Review · Reviewer_ggKb · 2025-10-28

**Soundness:** 2
**Presentation:** 2
**Contribution:** 2
**Rating:** 6
**Confidence:** 4

**Summary:**

This paper introduces AutoExpert, a benchmark for automated data annotation based on expert-crafted annotation guidelines. The task mimics human annotators who learn to label LiDAR data with 3D bounding boxes using only a few visual examples (4-8 RGB images per class) and textual descriptions of object classes, with no 3D cuboid annotations in the guidelines. The paper also propose methods for the new benchmark including Multi-Modal Few-Shot Finetuning and 3D Cuboid Generation via MHT.

Experiments show that the proposed method achieves 8.9×-11.6× speedup over grid search baseline, improves the mAP3D from 12.1 → 21.9 through progressive refinements and outperforms existing baselines (EMMS, LogME, LEEP, NLEEP, CM3D).

**Strengths:**

1. This paper is the first one to propose a benchmark for automated data annotation based on expert-crafted annotation guidelines.

2. The paper also propose Multi-Modal Few-Shot Finetuning and 3D Cuboid Generation via MHT for the benchmark.

**Weaknesses:**

Concern 1: Clarifying the Paper's Primary Contribution. The paper presents both a new task formulation and a solution pipeline, which creates some ambiguity about its primary contribution:

As a benchmark contribution: The work thoughtfully repurposes nuScenes with its annotation guidelines. However, expanding to additional datasets could strengthen generalizability claims.

As a methods contribution: The paper propose method including Multi-Modal Few-Shot Finetuning and 3D Cuboid Generation via MHT to lift the predicted 2D boxes to 3D. However, there exists literature lifting off-the-shelf 2D object detector to 3D [1,2,3], which the authors did not talk about and compare.

[1] Wang, Zitian, et al. "Object as query: Lifting any 2d object detector to 3d detection." Proceedings of the IEEE/CVF International Conference on Computer Vision. 2023.
[2] Ji, Haoxuanye, Pengpeng Liang, and Erkang Cheng. "Enhancing 3d object detection with 2d detection-guided query anchors." Proceedings of the IEEE/CVF Conference on Computer Vision and Pattern Recognition. 2024.
[3] Yang, Yung-Hsu, et al. "3D-MOOD: Lifting 2D to 3D for Monocular Open-Set Object Detection." Proceedings of the IEEE/CVF International Conference on Computer Vision. 2025.

Concern 2: Since the absolute performance is not high, the performance gain might come from random seed selection or training instability.

**Questions:**

1. Could the author discuss more on the literature that lifts off-the-shelf 2D object detector to 3D and provide experiments comparing to those methods?

2. The reviewer is wondering whether the experiments in the paper are conducted using the same random seed. Or maybe the authors can conduct repeated experiments to demonstrate the robustness of the proposed method.

---

### Official Review · Reviewer_QKjY · 2025-10-31

**Soundness:** 1
**Presentation:** 1
**Contribution:** 1
**Rating:** 2
**Confidence:** 5

**Summary:**

This paper introduces AutoExpert, a benchmark for auto-annotation from expert-crafted guidelines in autonomous driving. It repurposes nuScenes and tasks models to read textual rules plus a few 2D examples, then produce 3D LiDAR cuboids without any 3D labels. The authors propose a simple pipeline that adapts foundation models for 2D detection/segmentation and lifts detections to 3D via frustum reasoning, MHT search, sweep aggregation, and geometric/tracking-based scoring. Results show meaningful gains from this pipeline but a large gap to supervised 3D detectors, motivating LiDAR-based foundation models.

**Strengths:**

• Less efforts at the very early stage of labeling: The benchmark utilizes existing foundation models and lifting techniques to label 3D objects.
• Comprehensive Ablations: This paper conducts comprehensive ablation studies on foundation-model adaptation for 2D detection and on 3D lifting components.

**Weaknesses:**

1. [Limited Novelty] Using foundation models for auto-labeling is not new. Lifting 2D to 3D is an outdated method. Auto-labeling 3D objects is a very old and mature field.
2. [Questionable Results] The proposed method cannot be used for auto-labeling due to its poor performance. Existing lidar-based 3D object detector works much better than the proposed method. In addition, in Table 5, among several refinement factors (center, size, orientation, score), the size-only variant achieves the highest NDS, but the paper does not explain why size dominates or how NDS ends up peaking there. Please report a breakdown by error components (e.g., mATE, mASE, mAOE) and per-class analyses to justify this outcome.
3. [Presentation and Writing] The writing in this manuscript is very awkward and is not very legible and confusing. The overall presentation also needs significant improvements.
4. [Reproducibility of prompt] Prompt performance can be sensitive to the search-space design and API settings, yet the paper does not specify these nor report variance/confidence intervals from repeated runs. Table 1 presents a single-run result only, so robustness to different seeds/settings is unclear. Please report the full specification (e.g., search space, API parameters, seed control) and release the exact prompts used to ensure reproducibility.

**Questions:**

1. What is the computational overhead of the 3D lifting stage (MHT search, sweep aggregation, scoring, and tracking) per frame?
2. What data volume is required for the 3D refinement network to yield meaningful NDS improvement?
3. What failure modes dominate (e.g., occlusion, glass reflections, fences), and where do SA/S₃D/track help most?

---

### Official Review · Reviewer_hYFb · 2025-11-01

**Soundness:** 2
**Presentation:** 3
**Contribution:** 2
**Rating:** 2
**Confidence:** 4

**Summary:**

This paper introduces AutoExpert, a novel benchmark for auto-annotation from expert-crafted guidelines. The whole pipeline is tested in 3D LiDAR detection tasks with the nuScenes dataset. The benchmark leverages human defined annotation rules and foundation model capabilities. By asking models to automatically annotate 3D cuboids solely from textual and visual guidelines, this paper realizes the annotation without any labeled LiDAR data. More specifically, the authors leverage GroundingDINO for 2D detection and SAM for object segmentation. They combine with 2D-to-3D lifting and Multi-Hypothesis Testing, and progressively refine key modules to improve performance.

**Strengths:**

- The presentation of this paper is smooth and clear, thus the paper is easy to read and understand.
- The paper provides a simple and clear pipeline and the code is attached. In the experiments, they provide evaluation results across different metrics (mAP2D/3D, NDS). Meanwhile, they have done comprehensive ablations, especially on sweep aggregation, geometric scoring, and tracking-based refinement.
- The paper considers both language understanding and cross-modal grounding in 3D LiDAR perception tasks. This could inspire research toward multi-modal LiDAR-based foundation models.

**Weaknesses:**

- While the paper consider the problem as auto-annotation from expert-crafted guidelines, the core idea largely overlaps with recent works in open-vocabulary LiDAR object detection [1,2,3]. These works have demonstrated similar strategies for bridging text and point clouds via intermediate images or cross-modal projection modules. These methods use 2D vision-language models (e.g., GroundingDINO, CLIP) to extract semantic cues and then lift them to 3D space, followed by geometric association or frustum-based fusion. These are essentially the same process adopted here.

- The “AutoExpert” pipeline mainly adopts existing components (GroundingDINO + SAM + GPT-4o prompting + frustum clustering + MHT). Thus, the pipeline does not introduce a fundamentally new mechanism for linking textual guidelines and 3D data. From the perspective of system engineering, the paper is ok, however, it is relatively not enough since open-vocabulary detection works already have the same pipeline.

- Only one nuScenes dataset is considered for 3D detection benchmark. To validate “Auto-annotation from expert guidelines,” more comprehensively, one would expect at least one additional dataset considering the generalization ability.

- Works such as PointLLM, LidarGPT, or LLaVA-3D are not discussed or compared.

- The paper could be much more solid if it shows some examples where 2D-to-3D lifting fails, e.g., failure cases caused by depth ambiguity, occlusion, and semantic inaccuracy in text prompts.

[1] Open‑Vocabulary Point‑Cloud Object Detection Without 3D Annotation

[2] OpenSight: A Simple Open‑Vocabulary Framework for LiDAR‑Based Object Detection

[3] Find n' Propagate: Open‑Vocabulary 3D Object Detection in Urban Environments

**Questions:**

- As I mentioned above, the core contribution and novelty of this paper could be more clear, compared to existing works.

- Investigate uncertainty-aware techniques to quantify ambiguity from 2D detections could yield more informative pseudo-labels.

- Add multiple datasets evaluation or synthetic guideline experiments to show scalability beyond nuScenes.

---

### Official Review · Reviewer_Y8dz · 2025-11-03

**Soundness:** 3
**Presentation:** 3
**Contribution:** 2
**Rating:** 4
**Confidence:** 3

**Summary:**

This paper proposes a new benchmark named AutoExpert for Auto-Annotation from Expert-Crafted Guidelines. This work aims to generate 3D LiDAR cuboids from the expert-crafted guideline text and a few class images without 3D label. The authors build up a modular pipeline that (1) detects and segments objects through foundation models, (2) then projects 2D detections into 3D, and (3) lifts to 3D via a simple multi-hypothesis testing (MHT) cuboid fitting with geometry score, sweep aggregation, and temporal score refinement. Experimental results show that 3D detection mAP improved from 12.1 to 21.9.

**Strengths:**

1. The paper provides clear motivation and is well organized; the figures are readable and understandable.
2. This work establishes a new and realistic benchmark for how labeling actually happens and fills the gap between self-supervision and supervised labeling.
3. The experimental results analysis is clear and supports authors’ claims.

**Weaknesses:**

1. The contribution of algorithm novelty is modest. The core components, including GroundingDINO, SAM, geometric lifting, and MHT, are standard. The main contribution is the new benchmark.

2. The evaluation was only conducted on one dataset; the generality of the guidelines-only setting remains unvalidated beyond nuScnens,  which may weaken the authors' claims.

3. Dependence on proprietary FMs (for synonyms/priors and 2D proposals) may impede comparability and reproducibility across groups without the same access.

**Questions:**

1. Can you train a standard 3D detector solely on auto-labels and report the proportion of human labels needed to match a supervised baseline (label-efficiency curve)?
2. How do mAP/NDS and per-class AP change when replacing GPT-4-class models with open-source LLM/VLMs? Please provide cost-adjusted comparisons.
3. Could a lightweight learned refiner replace MHT and improve accuracy-compute trade-offs? Please discuss failure modes where MHT systematically errs (e.g., yaw snaps, over-tight boxes).

---

### Note · Authors · 2025-11-12

I have read and agree with the venue's withdrawal policy on behalf of myself and my co-authors.